# An updated phylogeny of the *Alphaproteobacteria* reveals that the parasitic *Rickettsiales* and *Holosporales* have independent origins

Sergio A Muñoz-Gómez[1,2], Sebastian Hess[1,2,3], Gertraud Burger[4], B Franz Lang[4], Edward Susko[2,5], Claudio H Slamovits[1,2], Andrew J Roger[1,2]*

[1]Department of Biochemistry and Molecular Biology, Dalhousie University, Halifax, Canada; [2]Centre for Comparative Genomics and Evolutionary Bioinformatics, Dalhousie University, Halifax, Canada; [3]Institute of Zoology, University of Cologne, Cologne, Germany; [4]Department of Biochemistry, Robert-Cedergren Center in Bioinformatics and Genomics, Université de Montréal, Montreal, Canada; [5]Department of Mathematics and Statistics, Dalhousie University, Halifax, Canada

**Abstract** The *Alphaproteobacteria* is an extraordinarily diverse and ancient group of bacteria. Previous attempts to infer its deep phylogeny have been plagued with methodological artefacts. To overcome this, we analyzed a dataset of 200 single-copy and conserved genes and employed diverse strategies to reduce compositional artefacts. Such strategies include using novel dataset-specific profile mixture models and recoding schemes, and removing sites, genes and taxa that are compositionally biased. We show that the *Rickettsiales* and *Holosporales* (both groups of intracellular parasites of eukaryotes) are not sisters to each other, but instead, the *Holosporales* has a derived position within the *Rhodospirillales*. A synthesis of our results also leads to an updated proposal for the higher-level taxonomy of the *Alphaproteobacteria*. Our robust consensus phylogeny will serve as a framework for future studies that aim to place mitochondria, and novel environmental diversity, within the *Alphaproteobacteria*.
DOI: https://doi.org/10.7554/eLife.42535.001

*For correspondence:
Andrew.Roger@Dal.Ca

Competing interests: The authors declare that no competing interests exist.

## Introduction

The *Alphaproteobacteria* is an extraordinarily diverse and disparate group of bacteria and well-known to most biologists for also encompassing the mitochondrial lineage (*Williams et al., 2007*; *Roger et al., 2017*). The *Alphaproteobacteria* has massively diversified since its origin, giving rise to, for example, some of the most abundant (e.g. *Pelagibacter ubique*) and metabolically versatile (e.g. *Rhodobacter sphaeroides*) cells on Earth (*Giovannoni, 2017*; *Madigan et al., 2009*). The basic structure of the tree of the *Alphaproteobacteria* has largely been inferred through the analyses of 16S rRNA genes and several conserved proteins (*Garrity et al., 2005*; *Lee et al., 2005*; *Rosenberg et al., 2014*; *Fitzpatrick et al., 2006*; *Williams et al., 2007*; *Brindefalk et al., 2011*; *Georgiades et al., 2011*; *Thrash et al., 2011*; *Luo, 2015*). Today, eight major orders are well recognized, namely the *Caulobacterales*, *Rhizobiales*, *Rhodobacterales*, *Pelagibacterales*, *Sphingomonadales*, *Rhodospirillales*, *Holosporales* and *Rickettsiales* (the latter two formerly grouped into the *Rickettsiales sensu lato*), and their interrelationships have also recently become better understood (*Viklund et al., 2012*; *Viklund et al., 2013*; *Rodríguez-Ezpeleta and Embley, 2012*; *Wang and Wu, 2014*). These eight orders were grouped into two subclasses by *Ferla et al. (2013)*: the subclass

**eLife digest** The *Alphaproteobacteria* form one of the most abundant groups of bacteria on Earth, and one that is closely linked to all complex forms of life. Many bacteria within this class live inside the cells of other organisms. For example, mitochondria – the powerhouses of animal, plant and other eukaryotic cells – evolved from bacteria within this group. Other alphaproteobacteria act as parasites or beneficial symbionts within cells.

The history of life on Earth can be thought of as a tree, with each branch representing the evolution of a new species from a common ancestor. But for many bacteria, the earliest stages of their evolutionary history are so tangled and complex that their origin remains largely unknown. For example, efforts to study the earliest history of the *Alphaproteobacteria* have been plagued with errors and artefacts. The extreme variation in the genetic sequences of different bacteria in the group make it particularly challenging to uncover relationships between the species.

To overcome this problem, Muñoz-Gómez et al. focused on a set of 200 genes that occur in all alphaproteobacteria, and used a range of strategies to reduce potential errors in the data. The results propose a new general structure for the evolutionary tree of the *Alphaproteobacteria*. This shows that two groups of alphaproteobacteria that were thought to be closely related to each other – the parasites *Rickettsiales* and *Holosporales* – are unrelated. Instead, these groups evolved independently from different free-living alphaproteobacteria.

The abundance and diversity of the *Alphaproteobacteria* means that the improved understanding of their evolutionary origins could influence the work of a wide range of scientists. Further research could help to shed light on how parasitic bacteria interact with the cells they invade; reveal how bacteria evolved certain abilities, such as the ability to photosynthesize; and uncover the precise origin of mitochondria.

DOI: https://doi.org/10.7554/eLife.42535.002

*Rickettsiidae* comprising the order *Rickettsiales* and *Pelagibacterales*, and the subclass *Caulobacteridae* comprising all other orders.

The great diversity of the *Alphaproteobacteria* itself presents a challenge to deciphering the deepest divergences within the group. Such diversity encompasses a broad spectrum of genome (nucleotide) and proteome (amino acid) compositions (e.g. the A + T%-rich *Pelagibacterales versus* the G + C%-rich *Acetobacteraceae*) and molecular evolutionary rates (e.g. the fast-evolving *Pelagibacterales*, *Rickettsiales* or *Holosporales versus* many slow-evolving species in the *Rhodospirillales*) (*Ettema and Andersson, 2009*). This diversity may lead to pervasive artefacts when inferring the phylogeny of the *Alphaproteobacteria*, for example, long-branch attraction (LBA) between the *Rickettsiales* and *Pelagibacterales*, especially when including mitochondria (*Rodríguez-Ezpeleta and Embley, 2012*; *Viklund et al., 2012*; *Viklund et al., 2013*; *Luo, 2015*). Moreover, there are still important unknowns about the deep phylogeny of the *Alphaproteobacteria* (*Williams et al., 2007*; *Ferla et al., 2013*), for example, the divergence order among the *Rhizobiales*, *Rhodobacterales* and *Caulobacterales* (*Williams et al., 2007*), the monophyly of the *Pelagibacterales* (*Viklund et al., 2013*) and the *Rhodospirillales* (*Ferla et al., 2013*), and the precise placement of the *Rickettsiales* and its relationship to the *Holosporales* (*Wang and Wu, 2013*; *Martijn et al., 2018*).

Systematic errors stemming from using over-simplified evolutionary models (which often do not fit complex data as well by, for example, not accounting for compositional heterogeneity across sites or branches) are perhaps the major confounding and limiting factor to inferring deep evolutionary relationships; the number of taxa and genes (or sites) can also be important factors. Previous multi-gene tree studies of the *Alphaproteobacteria* were compromised by at least one of these problems, namely, simpler or less realistic evolutionary models (because they were not available at the time; for example, *Williams et al., 2007* used the simple WAG+Γ4 model that cannot account for compositional heterogeneity across sites), poor or uneven taxon sampling (because the focus was too narrow or few genomes were available; for example, *Williams et al., 2007* had very few rhodospirillaleans and no holosporaleans; *Georgiades et al., 2011* included only 42 alphaproteobacteria with only one pelagibacteralean) or a small number of genes (because the focus was mitochondria; for example, *Rodríguez-Ezpeleta and Embley, 2012* used 24 genes; *Wang and Wu, 2015*

relied on 29 genes; *Martijn et al., 2018* also used 24 genes; or because only a small set of 28 compositionally homogeneous genes was used, for example, *Luo, 2015*). The most recent study on the phylogeny of the *Alphaproteobacteria*, and mitochondria, attempted to counter systematic errors (or phylogenetic artefacts) by reducing amino acid compositional heterogeneity (*Martijn et al., 2018*). Even though some deep relationships were not robustly resolved, these analyses suggested that the *Pelagibacterales*, *Rickettsiales* and *Holosporales*, which have compositionally biased genomes, are not each other's closest relatives (*Martijn et al., 2018*). A resolved and robust phylogeny of the *Alphaproteobacteria* is fundamental to addressing questions such as how streamlined bacteria, intracellular parasitic bacteria, or mitochondria evolved from their alphaproteobacterial ancestors. Therefore, a systematic study of the different biases affecting the phylogeny of the *Alphaproteobacteria*, and its underlying data, is much needed.

Here, we revised the phylogeny of the *Alphaproteobacteria* by using a large dataset of 200 conserved single-copy genes and employing carefully designed strategies aimed at alleviating phylogenetic artefacts. We found that amino acid compositional heterogeneity, and more generally long-branch attraction, were major confounding factors in estimating phylogenies of the *Alphaproteobacteria*. In order to counter these biases, we used novel dataset-specific profile mixture models and recoding schemes (both specifically designed to ameliorate compositional heterogeneity), and removed sites, genes and taxa that were compositionally biased. We also present three draft genomes for endosymbiotic alphaproteobacteria belonging to the *Rickettsiales* and *Holosporales*: (1) an undescribed midichloriacean endosymbiont of *Peranema trichophorum*, (2) an undescribed rickettsiacean endosymbiont of *Stachyamoeba lipophora*, and (3) the holosporalean '*Candidatus* Finniella inopinata', an endosymbiont of the rhizarian amoeboflagellate *Viridiraptor invadens* (*Hess et al., 2016*). Our results provide the first strong evidence that the *Holosporales* is not closely related to the *Rickettsiales* and originated instead from within the *Rhodospirillales*. We incorporate these and other insights regarding the deep phylogeny of the *Alphaproteobacteria* into an updated taxonomy.

## Results

### The genomes and phylogenetic positions of three novel endosymbiotic alphaproteobacteria (*Rickettsiales* and *Holosporales*)

We sequenced the genomes of the novel holosporalean '*Candidatus* Finniella inopinata', an endosymbiont of the rhizarian amoeboflagellate *Viridiraptor invadens* (*Hess et al., 2016*), and two undescribed rickettsialeans, one associated with the heterolobosean amoeba *Stachyamoeba lipophora* and the other with the euglenoid flagellate *Peranema trichophorum*. The three genomes are small with a reduced gene number and high A + T% content, strongly suggesting an endosymbiotic

**Table 1.** Genome features for the three novel rickettsialeans sequenced in this study. See *Supplementary file 1* as well.

| Species | '*Candidatus* Finniella inopinata' | *Stachyamoeba*-associated rickettsialean | *Peranema*-associated rickettsialean |
|---|---|---|---|
| Genome size | 1,792,168 bp | 1,738,386 bp | 1,375,759 bp |
| N50 | 174,737 bp | 1,738,386 bp | 28,559 bp |
| Contig number | 28 | 1 | 125 |
| Gene number[†] | 1741 | 1588 | 1223 |
| A + T% content | 56.58% | 67.01% | 59.13% |
| Family | '*Candidatus* Paracaedibacteraeae' | *Rickettsiaceae* | '*Candidatus* Midichloriaceae' |
| Order | *Holosporales* | *Rickettsiales* | *Rickettsiales* |
| Completeness[‡] | 94.96% | 97.12% (=100%) | 92.08% |
| Redundancy[‡] | 0.0% | 0.0% | 2.1% |

[†]as predicted by Prokka v.1.13 (rRNA genes were searched with BLAST).
[‡]as estimated by Anvi'o v.2.4.0 using the *Campbell et al., 2013* marker gene set.
DOI: https://doi.org/10.7554/eLife.42535.003

lifestyle (*Table 1*). Comparisons of their rRNA genes show that these genomes are truly novel, being considerably divergent from other described alphaproteobacteria. As of February 2018, the closest 16S rRNA gene to that of the *Stachyamoeba*-associated rickettsialean belongs to *Rickettsia massiliae* str. AZT80, with only 88% identity. On the other hand, the closest 16S rRNA gene to that of the *Peranema*-associated rickettsialean belongs to an endosymbiont of *Acanthamoeba* sp. UWC8, which is only 92% identical. Phylogenetic analysis of both the 16S rRNA gene and a dataset that comprises 200 single-copy conserved marker genes (see below) confirm that each species belongs to different families and orders within the *Alphaproteobacteria* (*Supplementary file 1* and *Figure 2—figure supplement 1*). 'Candidatus Finniella inopinata' belongs to the recently described 'Candidatus Paracaedibacteraceae' in the *Holosporales* (*Hess et al., 2016*), whereas the *Stachyamoeba*-associated rickettsialean belongs to the *Rickettsiaceae*, and the *Peranema*-associated rickettsialean belongs to the 'Candidatus Midichloriaceae', in the *Rickettsiales*.

## Compositional heterogeneity appears to be a major confounding factor affecting phylogenetic inference of the *Alphaproteobacteria*

The average-linkage clustering of amino acid compositions shows that the *Rickettsiales*, *Pelagibacterales* (together with alphaproteobacterium HIMB59) and *Holosporales* are clearly distinct from other alphaproteobacteria. This indicates that these three taxa have divergent proteome amino acid compositions (*Figure 1A*). These taxa also have the lowest GARP:FIMNKY amino acid ratios in all the *Alphaproteobacteria* (*Figure 1A*; GARP amino acids are encoded by G + C%-rich codons, whereas FIMNKY amino acids are encoded by A + T%-rich codons. Proteomes that have low GARP:FIMNKY ratios are compositionally biased and therefore come from A + T%-rich genomes); the *Pelagibacterales* (including alphaproteobacterium HIMB59) being the most divergent, followed by the *Rickettsiales* and then the *Holosporales*. Such biased amino acid compositions appear to be the consequence of genome nucleotide compositions that are strongly biased toward high A + T%—a scatter plot of genome G + C% and proteome GARP:FIMNKY ratios shows a similar clustering of the *Rickettsiales*, *Pelagibacterales* (including alphaproteobacterium HIMB59) and *Holosporales* (*Figure 1B*). This compositional similarity in the proteomes of the *Rickettsiales*, *Pelagibacterales* (plus alphaproteobacterium HIMB59) and *Holosporales*, which also turn out to be the longest-branched alphaproteobacterial groups in previously published phylogenies (e.g. *Wang and Wu, 2015*), could be the outcome of either a shared evolutionary history (i.e. the groups are most closely

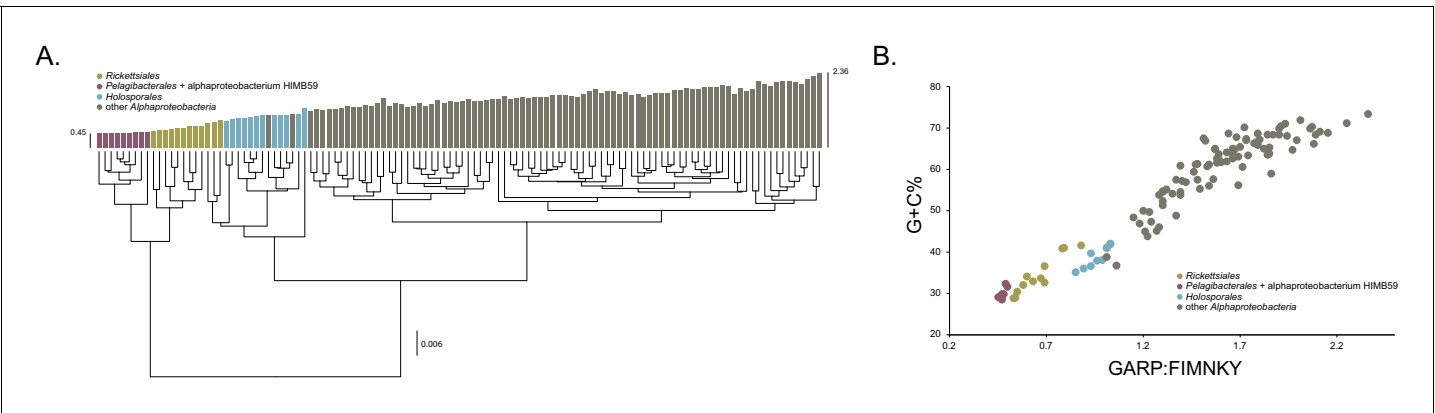

**Figure 1.** Compositional heterogeneity in the *Alphaproteobacteria* is a major factor that confounds phylogenetic inference. There are great disparities in the genome G + C% content and amino acid compositions of the *Rickettsiales*, *Pelagibacterales* (including alphaproteobacterium HIMB59) and *Holosporales* with all other alphaproteobacteria. (**A**) A UPGMA (average-linkage) clustering of amino acid compositions (based on the 200 gene set for the *Alphaproteobacteria*) shows that the *Rickettsiales* (brown), *Pelagibacterales* (maroon), and *Holosporales* (light blue) all have very similar proteome amino acid compositions. At the tips of the tree, GARP:FIMNKY amino acid ratio values are shown as bars. (**B**) A scatterplot depicting the strong correlation between G + C% (nucleotide compositions) and GARP:FIMNKY ratios (amino acid composition) for the 120 taxa in the *Alphaproteobacteria* (and outgroup) shows a similar clustering of the *Rickettsiales*, *Pelagibacterales* (including alphaproteobacterium HIMB59) and *Holosporales*.
DOI: https://doi.org/10.7554/eLife.42535.004

related to one another), or alternatively, evolutionary convergence (e.g. because of similar lifestyles or evolutionary trends toward small cell and genome sizes).

As a first step to discriminate between these two alternatives, we used maximum likelihood to estimate a tree on a dataset that comprised 200 single-copy and rarely laterally transferred marker genes for the *Alphaproteobacteria* (as determined by Phyla-AMPHORA; see Materials and methods for more details; *Wang and Wu, 2013*) under the site-heterogenous model LG+PMSF(ES60)+F+R6. The resulting tree united the *Rickettsiales*, *Pelagibacterales* (with alphaproteobacterium HIMB59 at its base) and *Holosporales* in a fully supported clade (*Figure 2A*; see *Figure 2—figure supplement 1* for labeled trees). The clustering of these three groups is suggestive of a phylogenetic artefact (e.g. long-branch attraction or LBA); indeed, such a pattern resembles the one seen in the tree of proteome amino acid compositions (see *Figure 1A*). This is because the three groups have the longest branches in the *Alphaproteobacteria* tree and have compositionally biased and fast-evolving genomes (see *Figure 2*). If evolutionary convergence in amino acid compositions is confounding phylogenetic inference for the *Alphaproteobacteria*, methods aimed at reducing compositional heterogeneity might disrupt the clustering of the *Rickettsiales*, *Pelagibacterales* and *Holosporales*.

To further test whether the clustering of the *Rickettsiales*, *Pelagibacterales* and *Holosporales* is real or artefactual, we used several different strategies to reduce the compositional heterogeneity of our dataset (see *Figure 2—figure supplement 2* for the diverse strategies employed). When removing the 50% most compositionally biased (heterogeneous) sites according to $\chi$ (a novel metric that measures amino acid compositional disparity at a site; see Materials and methods), the clustering between the *Rickettsiales*, *Pelagibacterales*, alphaproteobacterium HIMB59 and *Holosporales* is disrupted (*Figure 2B*; see also *Figure 2—figure supplement 3*). The new more derived placements for the *Pelagibacterales*, alphaproteobacterium HIMB59 and *Holosporales* are well supported (further described below), and support tends to increase as compositionally biased sites are removed (*Supplementary file 2A*). Furthermore, when each of these long-branched and compositionally biased taxa is analyzed in isolation (i.e. in the absence of the others), and compositional heterogeneity is further decreased, new phylogenetic patterns emerge that are incompatible, or in conflict, with their clustering (*Figure 2—figure supplement 4* and *Figure 3—figure supplements 1–5*). Various strategies to reduce compositional heterogeneity, such as removing the most compositionally biased sites, recoding the data into reduced character-state alphabets, or using only the most compositionally homogeneous genes, converge to very similar phylogenetic patterns for the *Alphaproteobacteria* in which the clustering of the *Rickettsiales*, *Pelagibacterales*, alphaproteobacterium HIMB59 and *Holosporales* is disrupted; the *Pelagibacterales*, alphaproteobacterium HIMB59 and *Holosporales* have much more derived phylogenetic placements (e.g., *Figure 3*, *Figure 2—figure supplement 4* and *Figure 3—figure supplements 1–5*). On the other hand, removing fast-evolving sites does not disrupt the clustering of these three long-branched groups (*Supplementary file 2B*), suggesting that high evolutionary rates per site are not a major confounding factor when inferring the phylogeny of the *Alphaproteobacteria*.

## The *Holosporales* is unrelated to the *Rickettsiales* and is instead most likely derived within the *Rhodospirillales*

The *Holosporales* has traditionally been considered part of the *Rickettsiales sensu lato* because it appears as sister to the *Rickettsiales* in many trees (e.g. *Hess et al., 2016*; *Montagna et al., 2013*; *Santos and Massard, 2014*). It is exclusively composed of endosymbiotic bacteria living within diverse eukaryotes, and such a lifestyle is shared with all other members of the *Rickettsiales* (with the possible exception of a recently reported ectosymbiotic rickettsialean; see *Castelli et al., 2018*). When we decrease, and then account for, compositional heterogeneity, we recover tree topologies in which the *Holosporales* moves away from the *Rickettsiales* (e.g. *Figure 2B*, *Figure 2—figure supplement 4B and D*). For example, the *Holosporales* becomes sister to all free-living alphaproteobacteria (the *Caulobacteridae*) when only the 40 most homogeneous genes are used (*Figure 2—figure supplement 4D*) or when 10% of the most compositionally biased sites are removed (*Supplementary file 2A*). When compositional heterogeneity is further decreased by removing 50% of the most compositionally biased sites, the *Holosporales* becomes sister to the *Rhodospirillales* (*Figure 2B* and *Supplementary file 2A*; and see also *Figure 2—figure supplement 4B*).

Similarly, when the long-branched and compositionally biased *Rickettsiales*, *Pelagibacterales*, and alphaproteobacterium HIMB59 (plus the extremely long-branched genera *Holospora* and

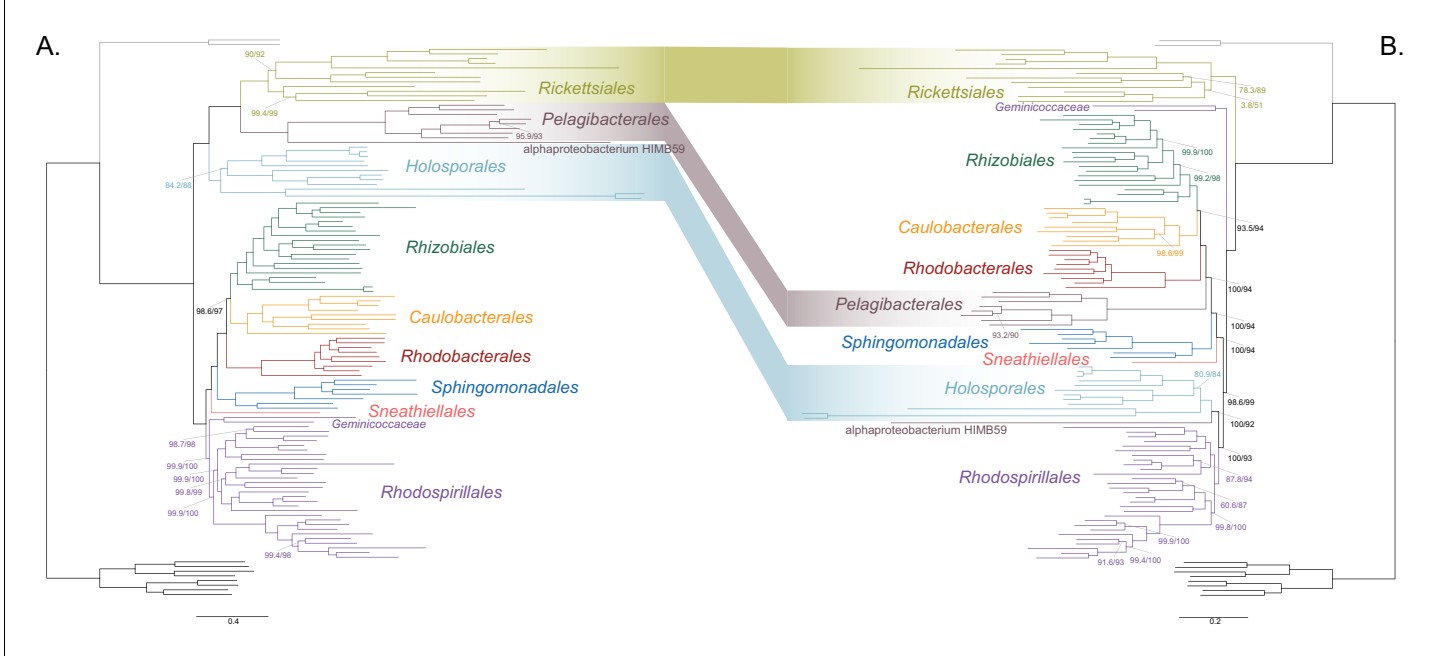

**Figure 2.** Decreasing compositional heterogeneity by removing compositionally biased sites disrupts the clustering of the *Rickettsiales*, *Pelagibacterales* (including alphaprotobacterium HIMB59) and *Holosporales*. All branch support values are 100% SH-aLRT and 100% UFBoot unless annotated. (**A**) A maximum-likelihood tree inferred under the LG + PMSF(ES60)+F + R6 model and from the untreated dataset which is highly compositionally heterogeneous. The three long-branched orders, the *Rickettsiales, Pelagibacterales* (including alphaproteobacterium HIMB59) and *Holosporales*, that have similar amino acid compositions form a clade. (**B**) A maximum-likelihood tree inferred under the LG + PMSF(ES60)+F + R6 model and from a dataset whose compositional heterogeneity has been decreased by removing 50% of the most biased sites according to $\chi$. In this phylogeny, the clustering of the *Rickettsiales, Pelagibacterales* and *Holosporales* is disrupted. The *Pelagibacterales* is sister to the *Rhodobacterales*, *Caulobacterales* and *Rhizobiales*. The *Holosporales*, and alphaproteobacterium HIMB59, become sister to the *Rhodospirillales*. The *Rickettsiales* remains as the sister to the *Caulobacteridae*. See *Figure 2—figure supplement 1* for taxon names. See *Figure 2—figure supplement 3* for the Bayesian consensus trees inferred in PhyloBayes MPI v1.7 under the CAT-Poisson+Γ4 model. See also *Figure 2—figure supplements 2* and *4–7*.
DOI: https://doi.org/10.7554/eLife.42535.005

The following figure supplements are available for figure 2:

**Figure supplement 1.** A labeled version showing taxon names for *Figure 2*.
DOI: https://doi.org/10.7554/eLife.42535.006

**Figure supplement 2.** A diagram of the strategies and phylogenetic analyses employed in this study.
DOI: https://doi.org/10.7554/eLife.42535.007

**Figure supplement 3.** Bayesian consensus trees inferred with PhyloBayes MPI v1.7 and the CAT-Poisson+Γ4 model.
DOI: https://doi.org/10.7554/eLife.42535.008

**Figure supplement 4.** Maximum-likelihood trees to assess the placements of the *Holosporales*, *Rickettsiales*, *Pelagibacterales* and alphaproteobacterium HIMB59 when all four groups are included.
DOI: https://doi.org/10.7554/eLife.42535.009

**Figure supplement 5.** Maximum-likelihood tree from the untreated dataset from which no taxon has been removed and analyzed under simpler LG4X model.
DOI: https://doi.org/10.7554/eLife.42535.010

**Figure supplement 6.** Constraint tree, used for IQ-TREE analyses, labeled with taxon names and also degree of missing data per taxon.
DOI: https://doi.org/10.7554/eLife.42535.011

**Figure supplement 7.** GARP:FIMNKY ratios across the proteomes of the 120 alphaproteobacteria and outgroup used in this study.
DOI: https://doi.org/10.7554/eLife.42535.012

'*Candidatus* Hepatobacter') are removed, after compositional heterogeneity had been decreased through site removal, the *Holosporales* move to a much more derived position well within the *Rhodospirillales* (*Figure 3A*, *Figure 3—figure supplement 1B and C* and *Figure 3—figure supplement 6*). If the very compositionally biased and fast-evolving *Holospora* and '*Candidatus* Hepatobacter' are left in, the *Holosporales* are pulled away from its derived position and the whole clade moves closer to the base of the tree (*Figure 3—figure supplement 7*). The same pattern in which the

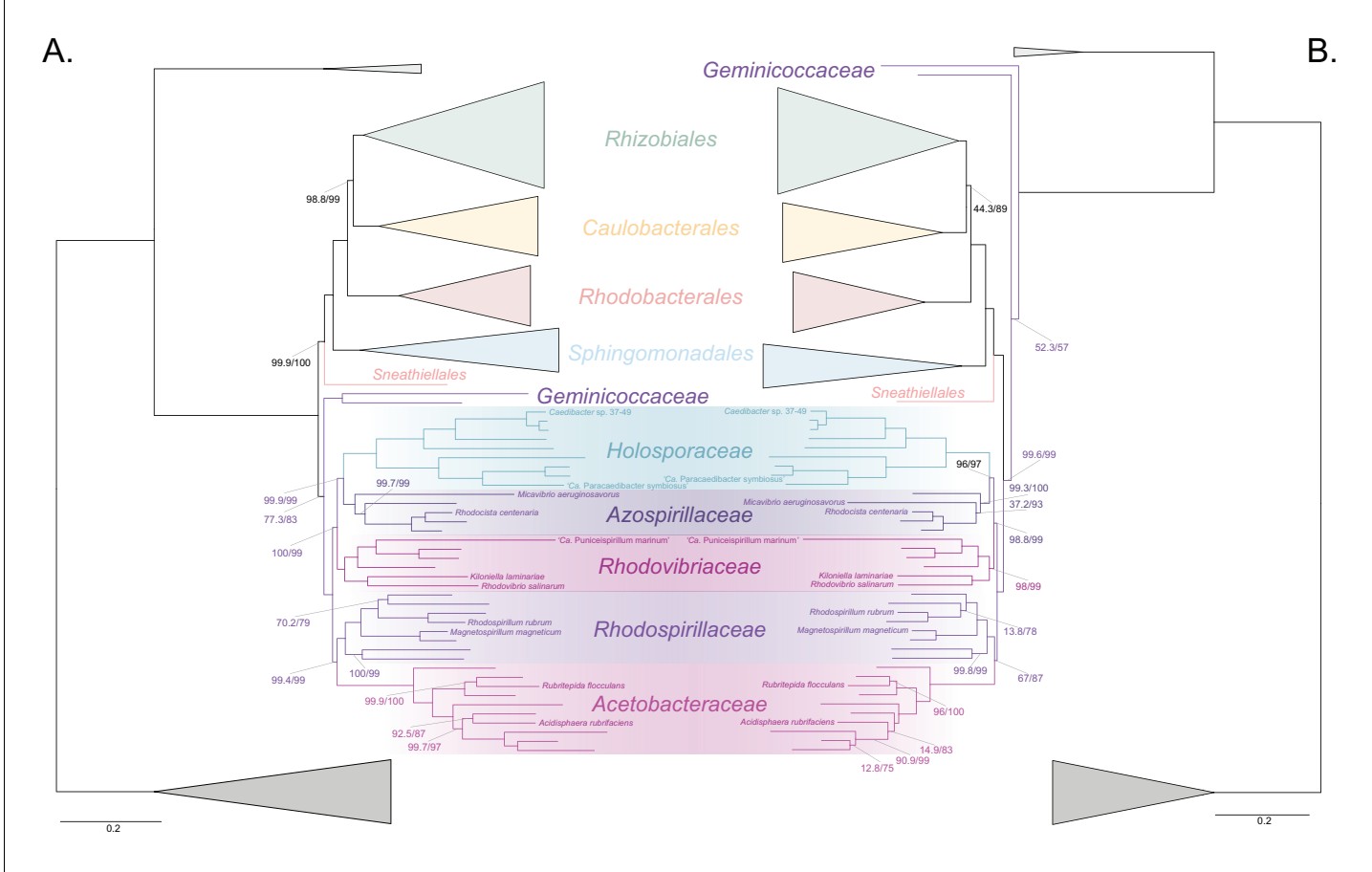

**Figure 3.** The *Holosporales* (renamed and lowered in rank to the *Holosporaceae* family here) branches in a derived position within the *Rhodospirillales* when compositional heterogeneity is reduced and the long-branched and compositionally biased *Rickettsiales*, *Pelagibacterales*, and alphaproteobacterium HIMB59 are removed. Branch support values are 100% SH-aLRT and 100% UFBoot unless annotated. (**A**) A maximum-likelihood tree, inferred under the LG + PMSF(ES60)+F + R6 model, to place the *Holosporaceae* in the absence of the *Rickettsiales*, *Pelagibacterales*, and alphaproteobacterium HIMB59 and when compositional heterogeneity has been decreased by removing 50% of the most biased sites. The *Holosporaceae* is sister to the *Azospirillaceae* fam. nov. within the *Rhodospirillales*. (**B**) A maximum-likelihood tree, inferred under the GTR + ES60 S4 +F + R6 model, to place the *Holosporaceae* in the absence of the *Rickettsiales*, *Pelagibacterales*, and alphaproteobacterium HIMB59, and when the data have been recoded into a four-character state alphabet (the dataset-specific recoding scheme S4: ARNDQEILKSTV GHY CMFP W) to reduce compositional heterogeneity. This phylogeny shows a pattern that matches that inferred when compositional heterogeneity has been alleviated through site removal. See *Figure 3—figure supplement 6* for the Bayesian consensus trees inferred in PhyloBayes MPI v1.7 and under the and the CAT-Poisson+Γ4 model. See also *Figure 3—figure supplements 1–5* and *7–8*.

DOI: https://doi.org/10.7554/eLife.42535.013

The following figure supplements are available for figure 3:

**Figure supplement 1.** Maximum-likelihood trees to assess the placement of the *Holosporales* in the absence of the *Rickettsiales*, *Pelagibacterales* and alphaproteobacterium HIMB59.

DOI: https://doi.org/10.7554/eLife.42535.014

**Figure supplement 2.** Maximum-likelihood trees to assess the placement of the *Rickettsiales* in the absence of the *Holosporales*, *Pelagibacterales*, and alphaproteobacterium HIMB59.

DOI: https://doi.org/10.7554/eLife.42535.015

**Figure supplement 3.** Maximum-likelihood trees to assess the placement of the *Rickettsiales* in the absence of the *Holosporales*, *Pelagibacterales*, alphaproteobacterium HIMB59 and the *Beta-*, and *Gammaproteobacteria* outgroup.

DOI: https://doi.org/10.7554/eLife.42535.016

**Figure supplement 4.** Maximum-likelihood trees to assess the placement of the *Pelagibacterales* in the absence of the *Holosporales*, *Rickettsiales* and alphaproteobacterium HIMB59.

DOI: https://doi.org/10.7554/eLife.42535.017

**Figure supplement 5.** Maximum-likelihood trees to assess the placement of alphaproteobacterium HIMB59 in the absence of the *Holosporales*, *Rickettsiales* and *Pelagibacterales*.

*Figure 3 continued on next page*

*Figure 3 continued*

DOI: https://doi.org/10.7554/eLife.42535.018

**Figure supplement 6.** Bayesian consensus trees inferred with PhyloBayes MPI v1.7 and the CAT-Poisson+Γ4 model.

DOI: https://doi.org/10.7554/eLife.42535.019

**Figure supplement 7.** Maximum-likelihood trees to assess the placement of the *Holosporales* when the fast-evolving *Holospora* and 'Candidatus Hepatobacter' are also included in the absence of the *Rickettsiales*, *Pelagibacterales* and alphaproteobacterium HIMB59.

DOI: https://doi.org/10.7554/eLife.42535.020

**Figure supplement 8.** Bayesian consensus tree inferred to place the *Holosporales* in the absence of the *Pelagibacterales*, alphaproteobacterium HIMB59, and *Rickettsiales*, and when the data have been recoded into a six-character state alphabet (the dataset-specific recoding scheme S6: AQEHISV RKMT PY DCLF NG W) to reduce compositional heterogeneity.

DOI: https://doi.org/10.7554/eLife.42535.021

*Holosporales* is derived within the *Rhodospirillales* is seen when these same taxa are removed, and the data are then recoded into four- or six-character states (*Figure 3B*, *Figure 3—figure supplement 6* and *Figure 3—figure supplement 8*). Specifically, the *Holosporales* now consistently branches as sister to a subgroup of rhodospirillaleans that includes, among others, the epibiotic predator *Micavibrio aeruginosavorus* and the purple nonsulfur bacterium *Rhodocista centenaria* (the *Azospirillaceae*, see below) (*Figure 3*). This new placement of the *Holosporales* has nearly full support under both maximum likelihood (>95% UFBoot; see *Figure 3*) and Bayesian inference (>0.95 posterior probability; see *Figure 3—figure supplement 6*). Thus, three different analyses independently converge to the same pattern and support a derived origin of the *Holosporales* within the *Rhodospirillales*: (1) removal of compositionally biased sites (*Figure 3A*), (2) data recoding into four-character states using the dataset-specific scheme S4 (*Figure 3B* and *Figure 3—figure supplement 7*), and (3) data recoding into six-character states using the dataset-specific scheme S6 (*Figure 3—figure supplement 8*); each of these strategies had to be combined with the removal of the *Pelagibacterales*, alphaproteobacterium HIMB59, and *Rickettsiales* to recover this phylogenetic position for the *Holosporales*.

A fourth independent analysis further supports a derived placement of the *Holosporales* nested within the *Rhodospirillales*. Bayesian inference using the CAT-Poisson+Γ4 model, on a dataset whose compositional heterogeneity had been decreased by removing 50% of the most compositionally biased sites but for which no taxon had been removed, also recovered the *Holosporales* as sister to the *Azospirillaceae* (see *Figure 2—figure supplement 3*).

## The *Rhodospirillales* is a diverse order and comprises five well-supported families

The *Rhodospirillales* is an ancient and highly diversified group, but unfortunately this is rarely obvious from published phylogenies because most studies only include a few species for this order (*Williams et al., 2007*; *Georgiades et al., 2011*; *Ferla et al., 2013*). We have included a total of 31 *Rhodospirillales* taxa to better cover its diversity. Such broad sampling reveals trees with five clear subgroups within the *Rhodospirillales* that are well-supported in most of our analyses (e.g. *Figures 2B* and *3*). First is the *Acetobacteraceae* which comprises acetic acid (e.g. *Acetobacter oboediens*), acidophilic (e.g. *Acidisphaera rubrifaciens*), and photosynthetic (bacteriochlorophyll-containing; for example, *Rubritepida flocculans*) bacteria. The *Acetobacteraceae* is strongly supported and relatively divergent from all other families within the *Rhodospirillales*. Sister to the *Acetobacteraceae* is another subgroup that comprises many photosynthetic bacteria, including the type species for the *Rhodospirillales*, *Rhodospirillum rubrum*, as well as the magnetotactic bacterial genera *Magnetospirillum*, *Magnetovibrio* and *Magnetospira* (*Figure 3*). This subgroup best corresponds to the poorly defined and paraphyletic *Rhodospirillaceae* family. We amend the *Rhodospirillaceae* taxon and restrict it to the clade most closely related to the *Acetobacteraceae*. As described above, when artefacts are accounted for, the *Holosporales* most likely branches within the *Rhodospirillales* and therefore we suggest the Holosporales sensu *Szokoli et al. (2016)* be lowered in rank to the family *Holosporaceae* (containing for example, *Caedibacter* sp. 37–49 and 'Candidatus Paracaedibacter symbiosus'), which is sister to the *Azospirillaceae* **fam. nov.** (*Figure 3*). The *Azospirillaceae* contains the purple bacterium *Rhodocista centenaria* and the epibiotic (neither periplasmic nor intracellular) predator *Micavibrio aeruginosavorus*, among others. The *Holosporaceae* and the *Azospirillaceae*

clades appear to be sister to the *Rhodovibriaceae* **fam. nov**. (*Figure 3*), a well-supported group that comprises the purple nonsulfur bacterium *Rhodovibrio salinarum*, the aerobic heterotroph *Kiloniella laminariae*, and the marine bacterioplankter 'Candidatus Puniceispirillum marinum' (or the SAR116 clade). Each of these subgroups and their interrelationships—with the exception of the *Holosporaceae* that branches within the *Rhodospirillales* only after compositional heterogeneity is countered— are strongly supported in nearly all of our analyses (e.g. see *Figures 2B* and *3*).

## The *Geminicoccaceae* might be sister to all other free-living alphaproteobacteria (the *Caulobacteridae*)

The *Geminicocacceae* is a recently proposed family within the *Rhodospirillales* (*Proença et al., 2018*). It is currently represented by only two genera, *Geminicoccus* and *Arboriscoccus* (*Foesel et al., 2007*; *Proença et al., 2018*). In most of our trees, however, *Tistrella mobilis* is often sister to *Geminicoccus roseus* with full statistical support (e.g., *Figures 2B* and *3A*, but see *Figure 3—figure supplement 6* for an exception) and we therefore consider it to be part of the *Geminococcaceae*. Interestingly, the *Geminicoccaceae* tends to have two alternative stable positions in our analyses, either as sister to all other families of the *Rhodospirillales* (e.g. *Figures 2A* and *3A*), or as sister to all other orders of the *Caulobacteridae* (i.e. representing the most basal lineage of free-living alphaproteobacteria; *Figure 2B* and *Figure 2—figure supplement 3B*, *Figure 3B* and *Figure 3—figure supplement 6*, or *Figure 2—figure supplement 4B*, *Figure 3—figure supplement 1C*, *Figure 3—figure supplement 2B–D*, *Figure 3—figure supplement 3B and D*, and *Figure 3—figure supplement 5C*). Our analyses designed to alleviate compositional heterogeneity, specifically site removal and recoding (without taxon removal), favor the latter position for the *Geminicoccaceae* (*Figures 2B* and *3B*). Moreover, as compositionally biased sites are progressively removed, support for the affiliation of the *Geminicoccaceae* with the *Rhodospirillales* decreases, and after 50% of the sites have been removed, the *Geminicoccaceae* emerges as sister to all other free-living alphaproteobacteria with strong support (>95% UFBoot; *Supplementary file 2A*). In further agreement with this trend, the much simpler model LG4X places the *Geminicocacceae* in a derived position as sister to the *Acetobacteraceae* (*Figure 2—figure supplement 5*), but as model complexity increases, and compositional heterogeneity is reduced, the *Geminicoccaceae* moves closer to the base of the *Alphaproteobacteria* (*Figures 2A* and *3A*). Such a placement suggests that the *Geminicoccaceae* may be a novel and independent order-level lineage in the *Alphaproteobacteria*. However, because of the uncertainty in our results we opt here for conservatively keeping the *Geminicoccaceae* as the sixth family of the *Rhodospirillales* (*Figure 3A*).

## Other deep relationships in the *Alphaproteobacteria* (*Pelagibacterales*, *Rickettsiales*, alphaproteobacterium HIMB59)

The clustering of the *Pelagibacterales* (formerly the SAR11 clade) with the *Rickettsiales* and *Holosporales* is more easily disrupted than that of the *Holosporales*, either when long-branched (or compositionally biased) taxon removal is performed to control for compositional attractions or not. The removal of compositionally biased sites (from 30% on; 16,320 out of 54,400 sites; see *Supplementary file 2A*, *Figure 2B*, *Figure 2—figure supplement 3B* and *Figure 3—figure supplement 4B*), data recoding into four-character states (*Figure 3—figure supplement 4C*), and a set of the most compositionally homogeneous genes (*Figure 3—figure supplement 4D*), all support a derived placement of the *Pelagibacterales* as sister to the *Rhodobacterales*, *Caulobacterales* and *Rhizobiales*. Attempts to account for compositional heterogeneity both across sites (e.g. *Rodríguez-Ezpeleta and Embley, 2012*; *Viklund et al., 2012*; *Viklund et al., 2013*; *Martijn et al., 2018*) and taxa (e.g. *Luo et al., 2013*; *Luo, 2015*) tend to disrupt the potentially artefactual clustering of the *Pelagibacterales* and the *Rickettsiales* (in contrast to the studies of for example, *Williams et al., 2007*; *Thrash et al., 2011*; *Georgiades et al., 2011*) that did not account for compositional heterogeneity). The *Caulobacterales* is sister to the *Rhizobiales*, and the *Rhodobacterales* sister to both (e.g. *Figures 2B* and *3*). This is consistent throughout most of our results and such interrelationships become very robustly supported as compositional heterogeneity is increasingly alleviated (*Supplementary file 2A*). The placement of the *Rickettsiales* as sister to the *Caulobacteridae* (i.e. all other alphaproteobacteria) remains stable across different analyses (see *Supplementary file 2A*, and also *Figure 2B* and *Figure 3—figure supplement 2*); this is also true when the other long-

branched taxa, the *Pelagibacterales*, alphaproteobacterium HIMB59 and Holosporales, and even the *Beta- Gammaproteobacteria* outgroup, are removed (see *Figure 3—figure supplement 2* and *Figure 3—figure supplement 3*). Yet, the interrelationships inside the *Rickettsiales* order remain uncertain; the 'Candidatus Midichloriaceae' becomes sister to the *Anaplasmataceae* when fast-evolving sites are removed (*Supplementary file 2B*), but to the *Rickettsiaceae* when compositionally biased sites are removed (*Supplementary file 2A*). The placement of alphaproteobacterium HIMB59 is uncertain (e.g. see *Figure 2* and *Figure 2—figure supplement 3*, and *Figure 2—figure supplement 4* and *Figure 3—figure supplement 5*; in contrast to *Grote et al., 2012*); taxon-removal analyses suggest that alphaproteobacterium HIMB59 is sister to the *Caulobacteridae* (*Figure 3—figure supplement 5*), but the inclusion of any other long-branched group immediately destabilizes this position (e.g. see *Figure 2* and *Figure 2—figure supplement 2*, and *Figure 2—figure supplement 4*). This is consistent with previous reports that suggest that alphaproteobacterium HIMB59 is not closely related to the *Pelagibacterales* (*Viklund et al., 2013*; *Martijn et al., 2018*).

## Discussion

We have employed a diverse set of strategies to investigate the phylogenetic signal contained within 200 genes for the *Alphaproteobacteria*. Specifically, such strategies were primarily aimed at reducing amino acid compositional heterogeneity among taxa—a phenomenon that permeates our dataset (*Figure 1*). Compositional heterogeneity is a clear violation of the phylogenetic models used in our, and previous, analyses, and known to cause phylogenetic artefacts (*Foster, 2004*). In the absence of more sophisticated models for inferring deep phylogeny (i.e. those that best fit complex data), the only way to counter artefacts caused by compositional heterogeneity is by removing compositionally biased sites or taxa, or recoding amino acids into reduced alphabets (e.g. see *Susko and Roger, 2007*; *Heiss et al., 2018*; *Viklund et al., 2012*). A combination of these strategies reveals that the *Rickettsiales sensu lato* (i.e. the *Rickettsiales* and *Holosporales*) is polyphyletic. Our analyses suggest that the *Holosporales* is derived within the *Rhodospirillales*, and that therefore this taxon should be lowered in rank and renamed the *Holosporaceae* family (see *Figures 2B* and *3*). The same methods suggest that the *Rhodospirillales* might indeed be a paraphyletic order and that the *Geminicoccaceae* could be a separate lineage that is sister to the *Caulobacteridae* (e.g. *Figure 2B*). These two results, combined with our broader sampling, reorganize the internal phylogenetic structure of the *Rhodospirillales* and show that its diversity can be grouped into at least five well-supported major families (*Figure 3*).

In 16S rRNA gene trees, the *Holosporales* has most often been allied to the *Rickettsiales* (*Montagna et al., 2013*; *Hess et al., 2016*). The apparent diversity of this group has quickly increased in recent years as more and more intracellular bacteria living within protists have been described (e.g. *Hess et al., 2016*; *Szokoli et al., 2016*; *Eschbach et al., 2009*; *Boscaro et al., 2013*). An endosymbiotic lifestyle is shared by all members of the *Holosporales* and is also shared with all those that belong to the *Rickettsiales*. Thus, it had been reasonable to accept their shared ancestry as suggested by some 16S rRNA gene trees (e.g. *Montagna et al., 2013*; *Santos and Massard, 2014*; *Hess et al., 2016*). Apparent strong support for the monophyly of the *Rickettsiales* and the *Holosporales* recently came from some multi-gene trees by *Wang and Wu (2014)*, and *Wang and Wu (2015)* who expanded sampling for the *Holosporales*. However, an alternative placement for the *Holosporales* as sister to the *Caulobacteridae* has been reported by *Ferla et al. (2013)* based on rRNA genes, by *Georgiades et al. (2011)* based on 65 genes, by Schulz et al., (2015) based on 139 genes, as well as by *Wang and Wu (2015)* based on 26, 29, or 200 genes (see the supplementary information in *Wang and Wu, 2015*). This placement was acknowledged by *Szokoli et al. (2016)*, who formally established the order *Holosporales*. Most recently, *Martijn et al. (2018)*, who used strategies to reduce compositional heterogeneity, and similarly to *Wang and Wu (2015)*, recovered a number of placements for the *Holosporales* within the *Alphaproteobacteria*; however, these different placements for the *Holosporales* were poorly supported. Here, we provide strong evidence for the hypothesis that the *Holosporales* is not related to the *Rickettsiales*, as suggested earlier (*Georgiades et al., 2011*; *Ferla et al., 2013*; *Szokoli et al., 2016*). The *Rickettsiales sensu lato* is polyphyletic. We show that the *Holosporales* is artefactually attracted to the *Rickettsiales* (e.g. *Figure 2A*), but as compositional bias is increasingly alleviated (through site removal and recoding), they move further away from them (*Figure 2B*). The *Holosporales* is placed within the

*Rhodopirillales* as sister to the family *Azospirillaceae* (*Figure 3*). The similar lifestyles of the *Holosporales* and *Rickettsiales*, as well as other features like the presence of an ATP/ADP translocase (*Wang and Wu, 2014*), are therefore likely the outcome of convergent evolution.

A derived origin of the *Holosporales* has important implications for understanding the origin of mitochondria and the nature of their ancestor. *Wang and Wu (2014)*, and *Wang and Wu (2015)* proposed that mitochondria are phylogenetically embedded within the *Rickettsiales sensu lato*. In their trees, mitochondria were sister to a clade formed by the *Rickettsiaceae*, *Anaplasmataceae* and 'Candidatus Midichloriaceae', and the *Holosporales* was itself sister to all of them. This phylogenetic placement for mitochondria suggested that the ancestor of mitochondria was an intracellular parasite (*Wang and Wu, 2014*). But if the *Holosporales* is a derived group of rhodospirillaleans as we show here (see *Figure 3*), then the argument that mitochondria necessarily evolved from parasitic alphaproteobacteria no longer holds. While the sisterhood of mitochondria and the *Rickettsiales sensu stricto* is still a possibility, such a relationship does not imply that the two groups shared a parasitic common ancestor (i.e. a parasitic ancestry for mitochondria). The most recent analyses of *Martijn et al. (2018)* suggest that mitochondria are sister to all known alphaproteobacteria, also suggesting their non-parasitic ancestry. Our study, and that of Martijn et al., thus complement each other and support the view that mitochondria most likely evolved from ancestral free-living alphaproteobacteria (*contra Sassera et al., 2011*; *Wang and Wu, 2014*; *Wang and Wu, 2015*).

The order *Rhodospirillales* is quite diverse and includes many purple nonsulfur bacteria as well as all magnetotactic bacteria within the *Alphaproteobacteria*. The *Rhodospirillales* is sister to all other orders in the *Caulobacteridae* and has historically been subdivided into two families: the *Rhodospirillaceae* and the *Acetobacteraceae*. Recently, a new family, the *Geminicoccaceae*, was established for the *Rhodospirillales* (*Proença et al., 2018*). However, some of our analyses suggest that the *Geminicoccaceae* might be sister to all other *Caulobacteridae* (e.g. *Figures 2B* and *3B*). This phylogenetic pattern, therefore, suggests that the *Rhodospirillales* may be a paraphyletic order. The placement of the *Geminicoccaceae* as sister to the *Caulobacteridae* needs to be further tested once more sequenced diversity for this group becomes available; if it were to be confirmed, the *Geminicoccaceae* should be elevated to the order level. Whereas the *Acetobacteraceae* is phylogenetically well-defined, there has been considerable uncertainty about the *Rhodospirillaceae* (e.g. *Ferla et al., 2013*), primarily because of poor sampling and a lack of resolution provided by the 16S rRNA gene. We subdivide the *Rhodospirillaceae sensu lato* into three subgroups (*Figure 3*). We restrict the *Rhodospirillaceae sensu stricto* to the subgroup that is sister to the *Acetobacteracae* (*Figure 3*). The other two subgroups are the *Rhodovibriaceae* and the *Azospirillaceae*; the latter is sister to the *Holosporaceae* (*Figure 3*).

Based on our fairly robust phylogenetic patterns, we have updated the higher-level taxonomy of the *Alphaproteobacteria* (*Table 2*). We exclude the *Magnetococcales* from the *Alphaproteobacteria* class because of its divergent nature (e.g. see Figure 1 in *Esser et al., 2007* which shows that many of *Magnetococcus*' genes are more similar to those of beta-, and gammaproteobacteria). In agreement with its intermediate phylogenetic placement, we endorse the *Magnetococcia* class as proposed by *Parks et al. (2018)*. At the highest level we define the *Alphaproteobacteria* class as comprising two subclasses *sensu Ferla et al. (2013)*, the *Rickettsidae* and the *Caulobacteridae*. The former contains the *Rickettsiales*, and the latter contains all other orders, which are primarily and ancestrally free-living alphaproteobacteria. The order *Rickettsiales* comprises three families as previously defined, the *Rickettsiaceae*, the *Anaplasmataceae*, and the 'Candidatus Midichloriaceae'. On the other hand, the *Caulobacteridae* is composed of seven phylogenetically well-supported orders: the *Rhodospirillales*, *Sneathiellales*, *Sphingomonadales*, *Pelagibacterales*, *Rhodobacterales*, *Caulobacterales* and *Rhizobiales*. Among the many species claimed to represent new order-level lineages on the basis of 16S rRNA gene trees (*Cho and Giovannoni, 2003*; *Kwon et al., 2005*; *Kurahashi et al., 2008*; *Wiese et al., 2009*; *Harbison et al., 2017*), only *Sneathiella* deserves order-level status (*Kurahashi et al., 2008*), since all others have derived placements in our trees and those published by others (*Williams et al., 2012*; *Bazylinski et al., 2013*; *Venkata Ramana et al., 2013*; *Harbison et al., 2017*). The *Rhodospirillales* order comprises six families, three of which are new, namely the *Holosporaceae*, *Azospirillaceae* and *Rhodovibriaceae* (*Table 2*). This new higher-level classification of the *Alphaproteobacteria* updates and expands those presented by *Ferla et al. (2013)*, the 'Bergey's Manual of Systematics of Archaea and Bacteria' (*Garrity et al., 2005*; *Whitman, 2015*), and 'The Prokaryotes' (*Rosenberg et al., 2014*). The classification scheme proposed

**Table 2.** A higher-level classification scheme for the *Alphaproteobacteria* and the *Magnetococcia* classes within the *Proteobacteria*, and the *Rickettsiales* and *Rhodospirillales* orders within the *Alphaproteobacteria*.

| |
|---|
| Class 1. *Alphaproteobacteria* **Garrity et al., 2005** |
| Subclass 1. *Rickettsidae* **Ferla et al., 2013** emend. Muñoz-Gómez et al. 2019 (this work) |
| Order 1. *Rickettsiales* **Gieszczkiewicz, 1939** emend. **Dumler et al., 2001** |
| Family 1. *Anaplasmataceae* **Philip, 1957**<br>Family 2. '*Candidatus* Midichloriaceae' **Montagna et al., 2013**<br>Family 3. *Rickettsiaceae* **Pinkerton, 1936** |
| Subclass 2. *Caulobacteridae* **Ferla et al., 2013** emend. Muñoz-Gómez et al. 2019 |
| Order 1. *Rhodospirillales* **Pfennig and Trüper, 1971** emend. Muñoz-Gómez et al. 2019 |
| Family 1. *Acetobacteraceae* (ex Henrici 1939) **Gillis and De Ley, 1980**<br>Family 2. *Rhodospirillaceae* **Pfennig and Trüper, 1971** emend. Muñoz-Gómez et al. 2019<br>Family 3. *Azospirillaceae* fam. nov. Muñoz-Gómez et al. 2019<br>Family 4. *Holosporaceae* **Szokoli et al., 2016**<br>Family 5. *Rhodovibriaceae* fam. nov. Muñoz-Gómez et al. 2019<br>Family 6. *Geminicoccaceae* **Proença et al., 2018** |
| Order 2. *Sneathiellales* **Kurahashi et al., 2008** |
| Order 3. *Sphingomonadales* **Yabuuchi and Kosako, 2005** |
| Order 4. *Pelagibacterales* **Grote et al., 2012** |
| Order 5. *Rhodobacterales* **Garrity et al., 2005** |
| Order 6. *Caulobacterales* **Henrici and Johnson, 1935** |
| Order 7. *Rhizobiales* **Kuykendall, 2005** |
| Class 2. *Magnetococcia* **Parks et al., 2018** |
| Order 1. *Magnetococcales* **Bazylinski et al., 2013** |

DOI: https://doi.org/10.7554/eLife.42535.022

here could be partly harmonized with that recently proposed by *Parks et al. (2018)* by elevating the six families within the *Rhodospirllales* to the order level; the trees by *Parks et al. (2018)*, however, are in conflict with those shown here and many of their proposed taxa are as well.

## Conclusions

We employed a combination of methods to decrease compositional heterogeneity in order to disrupt artefacts that arise when inferring the phylogeny of the *Alphaproteobacteria*. This is an example of the complex nature of the historical signal contained in modern genomes and the limitations of our current evolutionary models to capture these signals. A robust phylogeny of the *Alphaproteobacteria* is a precondition for placing the mitochondrial lineage. This is because including mitochondria certainly exacerbates the already strong biases in the data, and therefore represents additional sources of artefacts in phylogenetic inference (as seen in *Wang and Wu, 2015*) where the *Holosporales* is attracted by both mitochondria and the *Rickettsiales*). The robust phylogenetic framework developed here will serve as a reference for future studies that aim to place mitochondria and novel not-yet-cultured environmental diversity within the *Alphaproteobacteria*.

## Taxon descriptions

*Rickettsidae* emend. (*Alphaproteobacteria*) *Rickettsia* is the type genus of the subclass. The *Rickettsidae* subclass is here amended by redefining its circumscription so it remains monophyletic by excluding the *Pelagibacterales* order. The emended *Rickettsidae* subclass within the *Alphaproteobacteria* class is defined based on phylogenetic analyses of 200 genes which are predominantly single-copy and vertically inherited (unlikely laterally transferred) when compositional heterogeneity was decreased by site removal or recoding. Phylogenetic (node-based) definition: the least inclusive clade containing *Anaplasma phagocytophilum* HZ, *Rickettsia typhi* Wilmington, and '*Candidatus* Midichloria mitochondrii' IricVA. The *Rickettsidae* does not include: *Pelagibacter* sp. HIMB058, '*Candidatus* Pelagibacter sp.' IMCC9063, alphaproteobacterium HIMB59, *Caedibacter* sp. 37–49, '*Candidatus* Nucleicultrix amoebiphila' FS5, '*Candidatus* Finniella lucida', *Holospora obtusa* F1, *Sneathiella*

*glossodoripedis* JCM 23214, *Sphingomonas wittichii*, and *Brevundimonas subvibrioides* ATCC 15264.

*Caulobacteridae* emend. (*Alphaproteobacteria*) *Caulobacter* is the type genus of the subclass. The *Caulobacteridae* subclass is here amended by redefining its circumscription so it remains monophyletic by including the *Pelagibacterales* order. The emended *Caulobacteridae* subclass within the *Alphaproteobacteria* class is defined based on phylogenetic analyses of 200 genes which are predominantly single-copy and vertically inherited (unlikely laterally transferred) when compositional heterogeneity was decreased by site removal or recoding. Phylogenetic (node-based) definition: the least inclusive clade containing *Pelagibacter* sp. HIMB058, '*Candidatus* Pelagibacter sp.' IMCC9063, alphaproteobacterium HIMB59, *Caedibacter* sp. 37–49, '*Candidatus* Nucleicultrix amoebiphila' FS5, '*Candidatus* Finniella lucida', *Holospora obtusa* F1, *Sneathiella glossodoripedis* JCM 23214, *Sphingomonas wittichii*, and *Brevundimonas subvibrioides* ATCC 15264. The *Caulobacteridae* does not include: *Anaplasma phagocytophilum* HZ, *Rickettsia typhi* Wilmington, and '*Candidatus* Midichloria mitochondrii' IricVA.

*Azospirillaceae* fam. nov. (*Rhodospirillales*, *Alphaproteobacteria*) *Azospirillum* is the type genus of the family. This new family within the *Rhodospirillales* order is defined based on phylogenetic analyses of 200 genes which are predominantly single-copy and vertically inherited (unlikely laterally transferred). Phylogenetic (node-based) definition: the least inclusive clade containing *Micavibrio aeruginoavorus* ARL-13, *Rhodocista centenaria* SW, and *Inquilinus limosus* DSM 16000. The *Azospirillaceae* does not include: *Rhodovibrio salinarum* DSM 9154, '*Candidatus* Puniceispirillum marinum' IMCC 1322, *Rhodospirillum rubrum* ATCC 11170, *Terasakiella pusilla* DSM 6293, *Acidiphilium angustum* ATCC 49957, and *Elioraea tepidiphila* DSM 17972.

*Rhodovibriaceae* fam. nov. (*Rhodospirillales*, *Alphaproteobacteria*) *Rhodovibrio* is the type genus of the family. This new family within the *Rhodospirillales* order is defined based on phylogenetic analyses of 200 genes which are predominantly single-copy and vertically inherited (unlikely laterally transferred). Phylogenetic (node-based) definition: the least inclusive clade containing *Rhodovibrio salinarum* DSM 9154, *Kiloniella laminariae* DSM 19542, *Oceanibaculum indicum* P24, *Thalassobaculum salexigens* DSM 19539 and '*Candidatus* Puniceispirillum marinum' IMCC 1322. The *Rhodovobriaceae* does not include: *Rhodospirillum rubrum* ATCC 11170, *Terasakiella pusilla* DSM 6293, *Rhodocista centenaria* SW, *Micavibrio aeruginoavorus* ARL-13, *Acidiphilium angustum* ATCC 49957, and *Elioraea tepidiphila* DSM 17972.

*Rhodospirillaceae* emend. (*Rhodospirillales*, *Alphaproteobacteria*) *Rhodospirillum* is the type genus of the family. The *Rhodospirillaceae* family is here amended by redefining its circumscription so it remains monophyletic. The emended *Rhodospirillaceae* family within the *Rhodospirillales* order is defined based on phylogenetic analyses of 200 genes which are predominantly single-copy and vertically inherited (unlikely laterally transferred). Phylogenetic (node-based) definition: the least inclusive clade containing *Rhodospirillum rubrum* ATCC 11170, *Roseospirillum parvum* 930 l, *Magnetospirillum magneticum* AMB-1 and *Terasakiella pusilla* DSM 6293. The *Rhodospirillaceae* does not include: *Rhodocista centenaria* SW, *Micavibrio aeruginoavorus* ARL-13, '*Candidatus* Puniceispirillum marinum' IMCC 1322, *Rhodovibrio salinarum* DSM 9154, *Elioraea tepidiphila* DSM 17972, and *Acidiphilium angustum* ATCC 49957.

*Holosporaceae* (*Rhodospirillales*, *Alphaproteobacteria*) *Holospora* is the type genus of the family. The *Holosporaceae* family as defined here has the same taxon circumscription as the *Holosporales* order *sensu* Szokoli et al., 2016, but it is here lowered to the family level and placed within the *Rhodospirillales* order. The new family rank-level for this group is based on the phylogenetic analysis of 200 genes, which are predominantly single-copy and vertically inherited (unlikely laterally transferred), when compositional heterogeneity was decreased by site removal or recoding (and coupled to the removal of the long-branched taxa *Pelagibacterales* and *Rickettsiales*). The family contains three subfamilies (lowered in rank from a former family level) and one formally undescribed clade, namely, the *Holosporodeae*, and '*Candidatus* Paracaedibacteriodeae', '*Candidatus* Hepatincolodeae', and the *Caedibacter-Nucleicultrix* clade.

## Materials and methods

### Genome sequencing

Cultures of *Viridiraptor invadens* strain VirI02, the host of '*Candidatus* Finniella inopinata', were grown on the filamentous green alga *Zygnema pseudogedeanum* strain CCAC 0199 as described in *Hess and Melkonian (2013)*. Once the algal food was depleted, *Viridiraptor* cells were harvested by filtration through a cell strainer (mesh size 40 µm to remove algal cell walls) and centrifugation (~1000 g for 15 min). For short-read sequencing, DNA extraction of total gDNA was carried out with the ZR Fungal/Bacterial DNA MicroPrep Kit (Zymo Research) using a BIO101/Savant FastPrep FP120 high-speed bead beater and 20 µL of proteinase K (20 mg/mL). A sequencing library was made using the NEBNext Ultra II DNA Library Prep Kit (New England Biolabs). Paired-end DNA sequencing libraries were sequenced with an Illumina MiSeq instrument (Dalhousie University; Canada). (number of reads = 3,006,282, read length = 150 bp). For long-read sequencing, DNA extraction was performed using a CTAB and phenol-chloroform method. Total gDNA was further cleaned through a QIAGEN Genomic-Tip 20/G. A sequencing library was made using the Nanopore Ligation Sequencing Kit 1D (SQK-LSK108). Sequencing was done on a portable MinION instrument (Oxford Nanopore Technologies). (total bases = 191,942,801 bp, number of reads = 73,926, longest read = 32,236 bp, mean read length = 2,596 bp, mean read quality = 9.4).

*Peranema trichophorum* strain CCAP 1260/1B was obtained from the Culture Collection of Algae and Protozoa (CCAP, Oban, Scotland) and grown in liquid Knop media plus egg yolk crystals. Total gDNA was extracted following *Lang and Burger (2007)*. A paired-end sequencing library was made using a TruSeq DNA Library Prep Kit (Illumina). DNA sequencing libraries were sequenced with an Illumina MiSeq instrument (Genome Quebec Innovation Centre; Canada). (number of reads = 4,157,475, read length = 300 bp).

*Stachyamoeba lipophora* strain ATCC 50324 cells feeding on *Escherichia coli* were harvested and then broken up with pestle and mortar in the presence of glass beads (<450 µm diameter). Total gDNA was extracted using the QIAGEN Genomic G20 Kit. A paired-end sequencing library was made using a TruSeq DNA Library Prep Kit (Illumina). DNA sequencing libraries were sequenced with an Illumina MiSeq instrument (Genome Quebec Innovation Centre; Canada). (number of reads = 35,605,415, read length = 100 bp).

### Genome assembly and annotation

Short sequencing reads produced in an Illumina MiSeq from *Viridiraptor invadens*, *Peranema trichophorum*, and *Stachyamoeba lipophora* were first assessed with FASTQC v0.11.6 and then, based on its reports, trimmed with Trimmomatic v0.32 (*Bolger et al., 2014*) using the options: HEADCROP:16 LEADING:30 TRAILING:30 MINLEN:36. Illumina adapters were similarly removed with Trimmomatic v0.32 using the option ILLUMINACLIP. Long-sequencing reads produced in a Nanopore MinION instrument from *Viridiraptor invadens* were basecalled with Albacore v2.1.7, adapters were removed with Porechop v0.2.3, lambda phage reads were removed with NanoLyse v0.5.1, quality filtering was done with NanoFilt v2.0.0 (with the options '–headcrop 50 -q 8 l 1000'), and identity filtering against the high-quality short Illumina reads was done with Filtlong v0.2.0 (and the options '–keep_percent 90 –trim –split 500 –length_weight 10 min_length 1000'). Statistics were calculated throughout the read processing workflow with NanoStat v0.8.1 and NanoPlot v1.9.1. A hybrid co-assembly of both processed Illumina short reads and Nanopore long reads from *Viridiraptor invadens* was done with SPAdes v3.6.2 (*Bankevich et al., 2012*). Assemblies of the Illumina short reads from *Peranema trichophorum* and *Stachyamobea lipophora* were separately done with SPAdes v3.6.2 (*Bankevich et al., 2012*). The resulting assemblies for both *Viridiraptor invadens* and *Peranema trichophorum* were later separately processed with the Anvi'o v2.4.0 pipeline (*Eren et al., 2015*) and refined genome bins corresponding to '*Candidatus* Finniella inopinata' and the *Peranema*-associated rickettsialean were isolated primarily based on tetranucleotide sequence composition and taxonomic affiliation of its contigs. A single contig corresponding to the genome of the *Stachyamoeba*-associated rickettsialean was obtained from its assembly and this was circularized by collapsing the overlapping ends of the contig. Gene prediction and genome annotation was carried out with Prokka v.1.13 (see *Table 1*).

## Dataset assembly (taxon and gene selection)

The selection of 120 taxa was largely based on the phylogenetically diverse set of alphaproteobacteria determined by *Wang and Wu (2015)*. To this set of taxa, recently sequenced and divergent unaffiliated alphaproteobacteria were added, as well as those claimed to constitute novel order-level taxa. Some other groups, like the *Pelagibacterales*, *Rhodospirillales* and the *Holosporales*, were expanded to better represent their diversity. A set of four betaproteobacteria and four gammaproteobacterial were used as outgroup (see *Figure 2—figure supplement 6* for taxon names; see *Supplementary file 2C* for accession numbers).

A set of 200 gene markers (54,400 sites; 9.03% missing data, see *Figure 2—figure supplement 6*) defined by Phyla-AMPHORA was used (*Wang and Wu, 2013*). The genes are single-copy and predominantly vertically inherited as assessed by congruence among them (*Wang and Wu, 2013*). In brief, Phyla-AMPHORA searches for each marker gene using a profile Hidden Markov Model (HMM), then aligns the best hits to the profile HMM using hmmalign of the HMMER suite, and then trims the alignments using pre-computed quality scores (the mask) previously generated using the probabilistic masking program ZORRO (*Wu et al., 2012*; *Wang and Wu, 2013*). Phylogenetic trees for each marker gene were inferred from the trimmed multiple alignments in IQ-TREE v1.5.5 (*Minh et al., 2013*; *Nguyen et al., 2015*) and under the model LG4X + F model. Single-gene trees were examined individually to remove distant paralogues, contaminants or laterally transferred genes. All this was done before concatenating the single-gene alignments into a supermatrix with SequenceMatrix v 1.8 (*Vaidya et al., 2011*). Another smaller dataset of 40 compositionally homogenous genes (5570 sites; 5.98% missing data) was built by selecting the least compositionally heterogeneous genes from the larger 200 gene set according compositional homogeneity tests performed in P4 (*Foster, 2004*); see *Supplementary file 2D* for a list of the 40 most compositionally homogenous genes). This was done as an alternative way to overcome the strong compositional heterogeneity observed in datasets for the *Alphaproteobacteria* with a broad selection of taxa. In brief, the P4 tests rely on simulations based on a provided tree (here inferred for each gene under the model LG4X + F in IQ-TREE) and a model (LG + F + G4 available in P4) to obtain proper null distributions to which to compare the $X^2$ statistic. Most standard tests for compositional homogeneity (those that do not rely on simulate the data on a given tree) ignore correlation due to phylogenetic relatedness, and can suffer from a high probability of false negatives (*Foster, 2004*).

Variations of our full set were made to specifically assess the placement of each long-branched and compositionally biased group individually. In other words, each group with comparatively long branches (the *Rickettsiales*, *Pelagibacterales*, *Holosporales*, and alphaproteobacterium HIMB59) was analyzed in isolation, that is, in the absence of other long-branched and compositionally biased taxa. This was done with the purpose of reducing the potential artefactual attraction among these groups. Taxon removal was done in addition to compositionally biased site removal and data recoding into reduced character-state alphabets (for a summary of the different methodological strategies employed see *Figure 2—figure supplement 2*).

## Removal of compositionally biased and fast-evolving sites

As an effort to reduce artefacts in phylogenetic inference from our dataset (which might stem from extreme divergence in the evolution of the *Alphaproteobacteria*), we removed sites estimated to be highly compositionally heterogeneous or fast evolving. The compositional heterogeneity of a site was estimated by using a metric intended to measure the degree of disparity between the most % AT-rich taxa and all others. Taxa were ordered from lowest to highest proteome GARP:FIMNKY ratios; 'GARP' amino acids are encoded by %GC-rich codons, whereas 'FIMNKY' amino acids are encoded by %AT-rich codons. The resulting plot was visually inspected and a GARP:FIMNKY ratio cutoff of 1.06 (which represented a discontinuity or gap in the distribution which separated the long-branched and compositionally biased taxa *Pelagibacterales*, *Holosporales* and *Rickettsiales* from all others) was chosen to divide the dataset into low GARP:FMINKY (or %AT-rich) and higher GARP:FIMNKY (or 'GC-rich') taxa (*Figure 2—figure supplement 7*). Next, we determined the degree of compositional bias per site ($\chi$) for the frequencies of both FIMNKY and GARP amino acids between the %AT-rich and all other ('GC-rich') alphaproteobacteria. To calculate this metric for each site the following formula was used:

$$\zeta = (\pi\text{FIMNKY}_{\%\text{AT-rich}} - \pi\text{FIMNKY}_{\%\text{GC-rich}}) + (\pi\text{GARP}_{\%\text{GC-rich}} - \pi\text{GARP}_{\%\text{AT-rich}})$$

where $\pi FIMNKY$ and $\pi GARP$ are the sum of the frequencies for FIMNKY and GARP amino acids at a site, respectively, for either '% AT-rich' or '% GC-rich' taxa. According to this metric, higher values measure a greater disparity between %AT-rich alphaproteobacteria and all others; a measure of compositional heterogeneity or bias per site. The most compositionally heterogeneous sites according to $\zeta$ were progressively removed using the software SiteStripper (*Verbruggen, 2018*) in increments of 10%. We also progressively removed the fastest evolving sites in increments of 10%. Conditional mean site rates were estimated under the LG+C60+F+R6 model in IQ-TREE v1.5.5 using the '-wsr' flag (*Nguyen et al., 2015*).

## Data recoding

Our datasets were recoded into four- and six-character state amino acid alphabets using dataset-specific recoding schemes aimed at minimizing compositional heterogeneity in the data (*Susko and Roger, 2007*). The program minmax-chisq, which implements the methods of *Susko and Roger (2007)*, was used to find the best recoding schemes—please see *Figure 3*, *Figure 2—figure supplement 4* and *Figure 3—figure supplement 1–6*, and *Figure 3—figure supplement 8* legends for the specific recoding schemes used for each dataset. The approach uses the chi-squared ($X^2$) statistic for a test of homogeneity of frequencies as a criterion function for determining the best recoding schemes. Let $\pi_i$ denote the frequency of bin $i$ for the recoding scheme currently under consideration. For instance, suppose the amino acids were recoded into four bins: RNCM EHIPTWV ADQLKS GFY, then $\pi_4$ would be the frequency with which the amino acids G, F or Y were observed. Let $\pi_{is}$ be the frequency of bin $i$ for the $s$th taxa. Then the $X^2$ statistic for the null hypothesis that the frequencies are constant, over taxa, against the unrestricted hypothesis is

$$t_s = \sum_{is} (\pi_{is} - \pi_i)^2 / \pi_i$$

The $X^2$ statistic provides a measure of how different the frequencies for the $s$th taxa are from the average frequencies. The maximum $t_s$ over $s$ is taken as an overall measure of how heterogeneous the frequencies are for a given recoding scheme. The minmax-chisq program searches through recoding schemes, moving amino acids from one bin to another, to try to minimize the $maxt_s$ (*Susko and Roger, 2007*).

## Phylogenetic inference

The inference of phylogenies was primarily done under the maximum likelihood framework and using IQ-TREE v1.5.5 (*Minh et al., 2013*; *Nguyen et al., 2015*). ModelFinder in IQ-TREE v1.5.5 (*Kalyaanamoorthy et al., 2017*) was used to assess the best-fitting amino acid empirical matrix (e.g. JTT, WAG, and LG), on a maximum-likelihood tree, to our full dataset of 120 taxa and 200 conserved single-copy marker genes (see *Supplementary file 2E* and *Supplementary file 2F*). We first inferred guide trees (for a PMSF analysis) with a model that comprises the LG empirical matrix, with empirical frequencies estimated from the data (F), six rates for the FreeRate model to account for rate heterogeneity across sites (R6), and a mixture model with 60 amino acid profiles (C60) to account for compositional heterogeneity across sites—LG + C60+F + R6. Because the computational power and time required to properly explore the whole tree space (given such a big dataset and complex model) was too high, constrained tree searches were employed to obtain these initial guide trees (see *Figure 2—figure supplement 6* for the constraint tree). Many shallow nodes were constrained if they received maximum UFBoot and SH-aLRT support in a LG + PMSF(C60)+F + R6 analysis. All deep nodes, those relevant to the questions addressed here, were left unconstrained (*Figure 2—figure supplement 6*). The guide trees were then used together with a dataset-specific mixture model ES60 to estimate site-specific amino acid profiles, or a PMSF (Posterior Mean Site Frequency Profiles) model, that best account for compositional heterogeneity across sites (*Wang et al., 2018*). The dataset-specific empirical mixture model ES60 also has 60 categories but, unlike the general C60, was directly estimated from our large dataset of 200 genes and 120 alphaproteobacteria (and outgroup) using the methods described in *Susko et al. (2018)*; ModelFinder (*Kalyaanamoorthy et al., 2017*) suggests that the LG + ES60+F + R6 model is the best-fitting model; the R6 model

component, however, considerably increases computational burden; see *Supplementary file 2F* and *Supplementary file 2G*). Final trees were inferred using the LG + PMSF(ES60)+F + R6 model and a fully unconstrained tree search. Those datasets that produced the most novel topologies under maximum likelihood were further analyzed under a Bayesian framework using PhyloBayes MPI v1.7 and the CAT-Poisson+Γ4 model (*Lartillot and Philippe, 2004*; *Lartillot et al., 2009*). This model allows for a very large number of classes to account for compositional heterogeneity across sites and, unlike in the more complex CAT-GTR+Γ4 model, also allows for convergence to be more easily achieved between MCMC chains. PhyloBayes MCMC chains were run for at least 10,000 cycles until convergence between the chains was achieved and the largest discrepancy (i.e. maxdiff parameter) was ≤0.4 (except for the untreated dataset analyzed in *Figure 2—figure supplement 3A*; see *Supplementary file 2H* for several summary statistics for each PhyloBayes MCMC chain, including discrepancy and effective sample size values). A consensus tree was generated from two PhyloBayes MCMC chains using a burn-in of 500 trees and sub-sampling every 10 trees.

Phylogenetic analyses of recoded datasets into four-character state alphabets were analyzed using IQ-TREE v1.5.5 and the model GTR + ES60 S4+F + R6. ES60S4 is an adaptation of the dataset-specific empirical mixture model ES60 to four-character states. It is obtained by adding the frequencies of the amino acids that belong to each bin in the dataset-specific four-character state scheme S4 (see Data Recoding for details). Phylogenetics analyses of recoded datasets into six-character state alphabets were analyzed using PhyloBayes MPI v1.7 and the CAT-Poisson+Γ4 model. Maximum-likelihood analyses with a six-state recoding scheme could not be performed because IQ-TREE currently only supports amino acid datasets recoded into four-character states.

### Other analyses

The 16S rRNA genes of 'Candidatus Finniella inopinata', and the presumed endosymbionts of *Peranema trichophorum* and *Stachyamoeba lipophora* were identified with RNAmmer 1.2 server and BLAST searches. A set of 16S rRNA genes for diverse rickettsialeans and holosporaleans, and other alphaproteobacteria as outgroup, were retrieved from NCBI GenBank. The selection was based on *Hess and Melkonian (2013)*, *Szokoli et al. (2016)* and *Wang and Wu (2015)*. Environmental sequences for uncultured and undescribed rickettsialeans were retrieved by keeping the 50 best hits resulting from a BLAST search of our three novel 16S rRNA genes against the NCBI GenBank non-redundant (nr) database. The sequences were aligned with the SILVA aligner SINA v1.2.11 and all-gap sites were later removed. Phylogenetic analyses on this alignment were performed on IQ-TREE v1.5.5 using the GTR + F + R8 model.

A UPGMA (average-linkage) clustering of amino acid compositions based on the 200 gene set for the *Alphaproteobacteria* was built in MEGA 7 (*Kumar et al., 2016*) from a matrix of Euclidean distances between amino acid compositions of sequences exported from the phylogenetic software P4 (*Foster, 2004*; http://p4.nhm.ac.uk/index.html).

### Data availability

Sequencing data were deposited in NCBI GenBank under the BioProject PRJNA501864. The genomes of 'Candidatus Finniella inopinata', endosymbiont of *Peranema trichophorum* strain CCAP 1260/1B and endosymbiont of *Stachyamoeba lipophora* strain ATCC 50324 were deposited in NCBI GenBank under the accessions GCA_004210305.1, GCA_004210275.1 and GCA_003932735.1. Raw sequencing reads were deposited on the NCBI SRA archive under the accessions SRR8145469, SRR8145470, SRR8156519, SRR8156520, SRR8156521, SRR8156522. Multi-gene datasets as well as phylogenetic trees inferred in this study were deposited at Mendeley Data under the DOI: 10. 17632/75m68dxd83.2.

### Acknowledgements

Sergio A Muñoz-Gómez is supported by a Killam Predoctoral Scholarship and a Nova Scotia Graduate Scholarship. Sebastian Hess was supported by the German Research Foundation (DFG grant HE 7560/1-1). This work was supported by Natural Sciences and Engineering Research (NSERC) Discovery Grants 2016–06792 to AJR, RGPIN/05754–2015 to CHS, RGPIN/05286–2014 to GB, and RGPIN-2017–05411 to BFL. We thank Jon Jerlström Hultqvist and Gina Filloramo (both at Dalhousie University) for advice on long-read sequencing with the Nanopore MinION. We also thank Camilo A

Calderón-Acevedo for advice about taxonomic issues, and Franziska Szokoli for reading and commenting on a late version of this manuscript. Bruce Curtis (Dalhousie University) and Peter G Foster (Natural History Museum of London) kindly provided technical help with bioinformatics and with the software P4, respectively. We thank Joanny Roy, Georgette Kiethega, Matus Valach, and Shona Teijeiro (all at the Université de Montréal), and Drahomira Faktora (University of South Bohemia), for help with the culturing, DNA preparation and sequencing of the endosymbiont of *Peranema trichophorum*. Some of the genome data used in this study were produced by the US Department of Energy Joint Genome Institute (http://www.jgi.doe.gov/) in collaboration with the user community.

## Additional information

### Funding

| Funder | Grant reference number | Author |
| --- | --- | --- |
| Killam Trusts | | Sergio A Muñoz-Gómez |
| Deutsche Forschungsgemeinschaft | HE7560/1-1 | Sebastian Hess |
| Natural Sciences and Engineering Research Council of Canada | RGPIN/05286–2014 | Gertraud Burger |
| Natural Sciences and Engineering Research Council of Canada | RGPIN-2017–05411 | B Franz Lang |
| Natural Sciences and Engineering Research Council of Canada | RGPIN/05754–2015 | Claudio H Slamovits |
| Natural Sciences and Engineering Research Council of Canada | 2016–06792 | Andrew J Roger |

The funders had no role in study design, data collection and interpretation, or the decision to submit the work for publication.

### Author contributions

Sergio A Muñoz-Gómez, Conceptualization, Data curation, Formal analysis, Investigation, Methodology, Writing—original draft, Writing—review and editing; Sebastian Hess, Gertraud Burger, B Franz Lang, Resources, Writing—original draft; Edward Susko, Software, Methodology; Claudio H Slamovits, Conceptualization, Resources, Supervision, Funding acquisition, Investigation, Writing—original draft, Project administration, Writing—review and editing; Andrew J Roger, Conceptualization, Supervision, Funding acquisition, Writing—original draft, Project administration, Writing—review and editing

### Author ORCIDs

Sergio A Muñoz-Gómez  http://orcid.org/0000-0002-6200-474X
Andrew J Roger  http://orcid.org/0000-0003-1370-9820

### Decision letter and Author response

Decision letter https://doi.org/10.7554/eLife.42535.031
Author response https://doi.org/10.7554/eLife.42535.032

## Additional files

### Supplementary files

• Supplementary file 1. A 16S rRNA gene maximum-likelihood tree of the *Rickettsiales* and *Holosporales* that phylogenetically places the three endosymbionts whose genomes were sequenced in this study. (1) 'Candidatus Finniella inopinata' endosymbiont of *Viridiraptor invadens* strain Virl02, (2) an

alphaproteobacterium associated with *Peranema trichophorum* strain CCAP 1260/1B, and (3) an alphaproteobacterium associated with *Stachyamoeba lipophora* strain ATCC 50324. Branch support values are SH-aLRT and UFBoot.

DOI: https://doi.org/10.7554/eLife.42535.023

• Supplementary file 2. Supplementary tables. (**A**) Ultrafast bootstrap (UFBoot) variation for several clades discussed in this study as compositionally biased sites, according to $\chi$, are progressively removed in steps of 10%. (**B**) Ultrafast bootstrap (UFBoot) variation for several clades discussed in this study as the fastest sites are progressively removed in steps of 10%. (**C**) GenBank assembly accession numbers for the 120 alphaproteobacterial and outgroup genomes used in this study. (**D**) A list of the least compositionally heterogeneous genes out of the 200 single-copy and vertically inherited genes used in this study. (**E**) Model fit of amino acid replacement matrices as components of simple models that do not account for compositional heterogeneity across sites. Models are ordered from lowest to highest BIC. -LnL: log-likelihood; df: degrees of freedom or number of free parameters; AIC: Akaike information criterion; AICc: corrected Akaike information criterion; BIC: Bayesian information criterion. (**F**) Model fit of amino acid replacement matrices as components of complex models that account for compositional heterogeneity across sites. Models are ordered from lowest to highest BIC. -LnL: log-likelihood; df: degrees of freedom or number of free parameters; AIC: Akaike information criterion; AICc: corrected Akaike information criterion; BIC: Bayesian information criterion. (**G**) Model fit of LG + ES60+F for which the model component that accounts for rate heterogeneity across sites varies. Models are ordered from lowest to highest BIC. -LnL: log-likelihood; df: degrees of freedom or number of free parameters; AIC: Akaike information criterion; AICc: corrected Akaike information criterion; BIC: Bayesian information criterion. (**H**) Several summary statistics for the PhyloBayes MCMC chains run for each analysis under the CAT-Poisson+Γ4.

DOI: https://doi.org/10.7554/eLife.42535.024

• Transparent reporting form

DOI: https://doi.org/10.7554/eLife.42535.025

## Data availability

Sequencing data were deposited in NCBI GenBank under the BioProject PRJNA501864. The genomes of 'Candidatus Finniella inopinata', endosymbiont of *Peranema trichophorum* strain CCAP 1260/1B and endosymbiont of *Stachyamoeba lipophora* strain ATCC 50324 were deposited in NCBI GenBank under the accessions GCA_004210305.1, GCA_004210275.1 and GCA_003932735.1. Raw sequencing reads were deposited on the NCBI SRA archive under the accessions SRR8145469, SRR8145470, SRR8156519, SRR8156520, SRR8156521, SRR8156522. Multi-gene datasets as well as phylogenetic trees inferred in this study were deposited at Mendeley Data under the DOI: http://dx. doi.org/10.17632/75m68dxd83.2

The following datasets were generated:

| Author(s) | Year | Dataset title | Dataset URL | Database and Identifier |
|---|---|---|---|---|
| Muñoz-Gómez SA, Hess S, Burger G, Lang BF, Susko E, Slamovits CH, Roger AJ | 2018 | Sequence data from: An updated phylogeny of the *Alphaproteobacteria* reveals that the parasitic *Rickettsiales* and *Holosporales* have independent origins | https://www.ncbi.nlm. nih.gov/bioproject/ PRJNA501864 | NCBI BioProject, PRJNA501864 |
| Muñoz-Gómez SA, Hess S, Burger G, Lang BF, Susko E, Slamovits CH, Roger AJ | 2018 | Trees and datasets from: An updated phylogeny of the Alphaproteobacteria reveals that the parasitic Rickettsiales and Holosporales have independent origins | http://dx.doi.org/10. 17632/75m68dxd83.2 | Mendeley Data, 10. 17632/75m68dxd83.2 |

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
