## [Decision Letter]

Thank you for submitting your article "An updated phylogeny of the *Alphaproteobacteria* reveals that the *Rickettsiales* and *Holosporales* have independent origins" for consideration by *eLife*. Your article has been reviewed by three peer reviewers, one of whom is a member of our Board of Reviewing Editors, and the evaluation has been overseen by Patricia Wittkopp as the Senior Editor. The following individual involved in the review of your submission has agreed to reveal his identity: Iker Irisarri (Reviewer #2).

The reviewers have discussed the reviews with one another and the Reviewing Editor has drafted this decision to help you prepare a revised submission.

Summary:

Muñoz-Gómez et al. performed phylogenomic analyses to resolve evolutionary relationship of deeply branching lineages in *Alphaproteobacteria*. This is a challenging job, as different lineages within *Alphaproteobacteria* have very different genomic base and amino acid compositions, and also have very different evolutionary rates. The authors figured out that the compositional heterogeneity is the major factor that makes the estimate of *Alphaproteobacteria* phylogeny so difficult. They presented a number of new evolutionary relationships that differ from the previous results. Among these new observations, the most significant one is that *Rickettsiales* and *Holosporales* may have independent origins, which contradicts the well-established concept that these two intracellular symbiotic lineages shared a common ancestor which transited to intracellular lifestyle only once. The relative branching of the different alphaproteobacteria are tested with several sensitivity analyses, culminating in a consensus that is reflected into a new taxonomy.

Essential revisions:

1) While the new hypothesis of independent origins of the two intracellular lineages is interesting, this manuscript appears to have created more controversies of the evolutionary relationships among other important lineages in *Alphaproteobacteria* that have been discussed extensively in recent years. Perhaps the most important change is that *Pelagibacterales* becomes sister to the *Rhodobacterales, Caulobacterales* and *Rhizobiales* in the present study. *Pelagibacterales* takes a free-living lifestyle, but it shares a number of genomic and evolutionary traits (genome size, genomic GC content, and genomic evolutionary rate) with the above two intracellular lineages. This makes the phylogenetic placement of *Pelagibacterales* and the intracellular lineages in *Alphaproteobacteria* interesting but challenging. There have been several hypotheses for the phylogenetic placement of *Pelagibacterales*, most of which were proposed in studies that were not designed to resolve this phylogenetic controversy or did not use the correct evolutionary models to control for the intrinsic bias in the genome sequences. In the present study, to support their argument the authors only cited the results from the studies that show the similar phylogenetic placement of *Pelagibacterales*, but ignored other more relevant studies including a study published in 2015 (https://www.ncbi.nlm.nih.gov/pubmed/25431989) which provided the first conclusive evidence that compositional heterogeneity causes the difficulty in placing *Pelagibacterales* in the *Alphaproteobacteria* tree, based on which that paper was able to reject alternate hypotheses including the one that is shown in the present manuscript and the papers it cited.

We do not mean that the new hypothesis of independent origins of the two intracellular lineages must be wrong, but we think it is essential that, before reaching this exciting conclusion or proposing this attractive hypothesis, the authors should be able to repeat some of the important evolutionary relationships which have already had excellent progress (like the evolutionary position of *Pelagibacterales*, though it remains controversial) or should provide strong evidence against it if their new finding disagrees with it. Without this, any new significant proposal like the independent origins of the two intracellular lineages is not convincing.

2) There are a few places where the authors state results whose "data [are] not shown". The authors should either remove these statements or show the supporting data.

3) The authors test the position of long-branched lineages (*Rickettsiales, Holosporales, Pelagibacterales*) by removing two of them at a time. However, one hypothesis that was not tested is whether the position of *Rickettsiales* might be the product of a long-branch attraction to the distant outgroup. Please perform an analysis including *Rickettisales* (but not *Holosporales* and *Pelagibacterales*) and no outgroup and see if the position of *Rickettsiales* varies relative to the other lineages, which would suggest its position is the product of a long-branch attraction. Similarly, the reviewers were not convinced (or the text was not clear) by the conclusion that *Holosporales* are derived within *Rhodospirillales* (e.g. Discussion paragraph one, also shown in Figure 3). This totally depends on the position of *Alphaproteobacterium* HIMB 59, which is also unstable across the analyses. For example in Figure 2B, *Holosporales* + Alphaproteobacterium is the sister group of *Rhodospirillales* and not derived within. This is particularly important since the authors propose to lower the rank of *Holosporales* in their taxonomy.

4) Dataset assembly. The authors use the Wang and Wu, 2013, and complement it with recently sequenced species for completeness. But was this done using the Phyla-AMPHORA pipeline, or using another ad-hoc pipeline? First of all, it appears that the authors did not perform any kind of data curation to make sure that the new species did not include contaminations, deep paralogy or LGT issues, and in our opinion this is a must. Likewise, there is no information about the alignment algorithm and trimming (if performed).

5) Regarding phylogenetic analyses, it seems the LG replacement matrix was chosen without even comparing its better fit statistically (e.g. AIC or BIC). The authors use +R6 and +R8 to account for among-site rate heterogeneity in different analyses, but without an apparent reason for that. Lastly, please provide information on the ESS values for Bayesian runs to have a better grasp on the chain convergence.

6) One of aspects why the phylogeny of alphaproteobacteria is of broad interest is the mitochondrial lineage. We wonder why the authors did not try to place the mitochondria into their analyses. We assume this would bring additional biases into an already difficult dataset, but we think we could have gotten an interesting insight given the vast amount of analyses performed with various strategies to reduce systematic errors.

7) Abstract: it would be best to remove the fifth sentence, given that the support for these findings is not definitive. Additionally, it is important that you add that this study proposes an updated taxonomy for alphaproteobacteria, which is one of the major outcomes of your study.

8) Conclusions: please remove the last two sentences of the conclusion. The one before last could be said for every study ever done. The last sentence is a bigger topic but including a single sentence in the conclusions fails to do it justice. If you want to discuss this issue, please include a paragraph in the discussion – as is, it comes out of the blue (and it's not clear why phylogenetic inference will be improved; if additional sampling keeps adding long branches, it may very well be that more uncertainty is introduced).

Introduction, third paragraph: It is generally well accepted that these three factors (few taxa, few genes, and models with poor fit) lead to systematic error. But your claim that previous studies were compromised by one or more of these factors in this section seems very hand-wavy. Can you give specific examples? Simply saying taxon sampling / model usage was poor in this or that study seems subjective – please give specific information as to why these studies had suboptimal designs (e.g., how many taxa were included, which of the major groups were sampled, why the model was a poor fit, etc.)

Subsection “Compositional heterogeneity appears to be a major confounding factor affecting phylogenetic inference of the *Alphaproteobacteria*”, second paragraph: please briefly introduce in a short paragraph how you built the data matrix before you start describing how you analyzed it.

Subsection “The *Holosporales* is unrelated to the *Rickettsiales* and is instead most likely derived within the *Rhodospirillales*” and “The Geminicoccaceae might be basal to all other free-living alphaproteobacteria (the Caulobacteridae)”: there are no page (or supplement size) restrictions, so please show the data.

Figure 3: the figure lists taxonomy family names (e.g., Holosporales) but the legend discusses order family names (e.g., Holosporaceae) and what the triangles correspond to is not explained. Please clearly annotate the figure.

Figures 2 and 3: the color-coding scheme of the different clades doesn't appear consistent. Please revise.

---

## [Author Response]

Essential revisions:1) While the new hypothesis of independent origins of the two intracellular lineages is interesting, this manuscript appears to have created more controversies of the evolutionary relationships among other important lineages in Alphaproteobacteria that have been discussed extensively in recent years. Perhaps the most important change is that Pelagibacterales becomes sister to the Rhodobacterales, Caulobacterales and Rhizobiales in the present study. Pelagibacterales takes a free-living lifestyle, but it shares a number of genomic and evolutionary traits (genome size, genomic GC content, and genomic evolutionary rate) with the above two intracellular lineages. This makes the phylogenetic placement of Pelagibacterales and the intracellular lineages in Alphaproteobacteria interesting but challenging. There have been several hypotheses for the phylogenetic placement of Pelagibacterales, most of which were proposed in studies that were not designed to resolve this phylogenetic controversy or did not use the correct evolutionary models to control for the intrinsic bias in the genome sequences. In the present study, to support their argument the authors only cited the results from the studies that show the similar phylogenetic placement of Pelagibacterales, but ignored other more relevant studies including a study published in 2015 (https://www.ncbi.nlm.nih.gov/pubmed/25431989) which provided the first conclusive evidence that compositional heterogeneity causes the difficulty in placing Pelagibacterales in the Alphaproteobacteria tree, based on which that paper was able to reject alternate hypotheses including the one that is shown in the present manuscript and the papers it cited.

We have corrected this oversight. We now discuss the Luo, 2015, paper and its relevance in both the Introduction and the Results.

We do not mean that the new hypothesis of independent origins of the two intracellular lineages must be wrong, but we think it is essential that, before reaching this exciting conclusion or proposing this attractive hypothesis, the authors should be able to repeat some of the important evolutionary relationships which have already had excellent progress (like the evolutionary position of Pelagibacterales, though it remains controversial) or should provide strong evidence against it if their new finding disagrees with it. Without this, any new significant proposal like the independent origins of the two intracellular lineages is not convincing.

Several studies have suggested that the sisterhood between the *Pelagibacterales* and the *Rickettsiales* assumed by some (e.g., Williams et al., 2007, Georgiades et al., 2011, and Thrash et al., 2011) is most probably a phylogenetic artefact (e.g., Viklund et al., 2012, 2013, Rodriguez-Ezpeleta, 2012, Luo et al., 2013 and Luo, 2015). These studies have shown that when compositional heterogeneity across sites (e.g., Viklund et al., 2012, 2013 and Rodriguez-Ezpeleta, 2012) or taxa (e.g., Luo et al., 2013 and Luo 2015) is accounted for by using more complex evolutionary models, the *Pelagibacterales* branches away from the *Rickettsiales* and closer to other free-living alphaproteobacteria. The more basal placement for the *Pelagibacterales* found by Luo et al., 2013, and Luo, 2015, might be the consequence of some residual compositional attraction between the *Pelagibacterales* and *Rickettsiales* due to, e.g., (1) using of the Dayhoff recoding-scheme that is not designed specifically for reducing compositional heterogeneity, (2) applying the NDCH model to a set of both compositionally homogenous and heterogeneous genes, (3) applying the CAT-GTR model to a set of both compositionally homogenous and heterogeneous genes, and (4) not controlling for compositional attraction between taxa by selectively removing long-branched and compositionally biased taxa.

All the analyses in our study rely on models that account for compositional heterogeneity across sites (i.e., LG+PMSF(C60)+F+R6 in maximum-likelihood or CAT-Poisson in Bayesian analyses). We further compositionally homogenized our datasets by (1) removing the most compositionally biased sites, (2) recoding our datasets into four and six character-state recoding schemes that minimize compositional heterogeneity, or (3) using a set of the 40 least compositionally biased genes. Several of these independent strategies of phylogenetically analyzing the data converge into the same derived placement for the *Pelagibacterales* as sister to the *Rhizobiales, Caulobacterales* and *Rhodobacterales*. For example, removing the 30-70% most compositionally biased sites without removing any long-branched or compositionally biased taxon (i.e., the *Rickettsiales, Holosporales* and alphaproteobacterium sp. HIMB59) places the *Pelgibacterales* in its most derived position in both maximum-likelihood (Table S1 in Supplementary file 2 and Figure 2B or Figure 2—figure supplement 4B) and Bayesian analyses (Figure 2—figure supplement 3B). When the long-branched and compositionally biased *Rickettsiales, Holosporales* and alphaproteobacterium sp. HIMB59 are removed to diminish the chances of compositional attractions, the *Pelagibacterales* again branches in its most derived position as sister to the *Rhizobiales, Caulobacterales* and *Rhodobacterales* in three independent analyses: (1) when the most compositionally biased sites are removed (Figure 3—figure supplement 4B), (2) when the data is recoded into the four character-state recoding scheme S4 (Figure 3—figure supplement 4C), and (3) when a set of the 40 most compositionally homogenous genes is used (see Figure 3—figure supplement 4D).

In summary, multiple independent strategies in our own study, as well as three different studies published by others (Viklund et al., 2012, 2013 and Rodriguez-Ezpeleta, 2012), strongly support the view that the *Pelagibacterales* has a derived placement in the tree of the *Alphaproteobacteria* as sister to the *Rhizobiales, Caulobacterales* and *Rhodobacterales* orders. Please see subsection “Other deep relationships in the Alphaproteobacteria (Pelagibacterales, Rickettsiales, alphaproteobacterium sp. HIMB59)” for a new slightly expanded discussion on the placement of the *Pelagibacterales* relative to previous studies including those of Luo et al., 2013, and Luo, 2015, that disrupted the artefactual clustering of the *Pelagibacterales* and *Rickettsiales* by accounting for compositional heterogeneity across taxa.

2) There are a few places where the authors state results whose "data [are] not shown". The authors should either remove these statements or show the supporting data.

We have now added two new figures that show these data. Please see Figure 3—figure supplement 7 and Figure 2—figure supplement 5.

3) The authors test the position of long-branched lineages (Rickettsiales, Holosporales, Pelagibacterales) by removing two of them at a time. However, one hypothesis that was not tested is whether the position of Rickettsiales might be the product of a long-branch attraction to the distant outgroup. Please perform an analysis including Rickettisales (but not Holosporales and Pelagibacterales) and no outgroup and see if the position of Rickettsiales varies relative to the other lineages, which would suggest its position is the product of a long-branch attraction. Similarly, the reviewers were not convinced (or the text was not clear) by the conclusion that Holosporales are derived within Rhodospirillales (e.g. Discussion paragraph one, also shown in Figure 3). This totally depends on the position of Alphaproteobacterium HIMB 59, which is also unstable across the analyses. For example in Figure 2B, Holosporales + Alphaproteobacterium is the sister group of Rhodospirillales and not derived within. This is particularly important since the authors propose to lower the rank of Holosporales in their taxonomy.

We have performed additional phylogenetic analyses (e.g., removal of the most compositionally biased sites, recoding, a subset of the least compositionally biased genes) to place the *Rickettsiales* in which the outgroup (*Β*- and *Gammaproteobacteria*) and the three other long-branched lineages (i.e., the *Holosporales, Pelagibacterales* and alphaproteobacterium sp. HIMB59) have been removed. The results consistently agree among them and suggest that the *Rickettsiales* is sister to all other alphaproteobacteria. Please see Figure 3—figure supplement 3.

The placement of the *Holosporales* was assessed in several analyses in which the *Rickettsiales, Pelagibacterales* and also alphaproteobacterium sp. HIMB59 were removed (see Figure 3 and Figure 3—figure supplement 1). These results support the derived placement of the *Holosporales* (renamed as *Holosporaceae*) within the *Rhodospirillales*. Therefore, the placement of the *Holosporales* is not specifically dependent on alphaproteobacterium sp. HIMB59 but on the presence of other long-branched and compositionally biased groups in the dataset (e.g., see Figure 3—figure supplement 6). We have also now made it clear throughout the text that alphaproteobacteriuam sp. HIMB59 was selectively removed as a long-branched group when analyses that attempted to phylogenetically place the *Rickettsiales, Holosporales* or *Pelagibacterales* in the phylogeny of the *Alphaproteobacteria* were performed (e.g., see paragraph two of subsection “The Holosporales is unrelated to the *Rickettsiales* and is instead most likely derived within the Rhodospirillales”).

4) Dataset assembly. The authors use the Wang and Wu, 2013 and complement it with recently sequenced species for completeness. But was this done using the Phyla-AMPHORA pipeline, or using another ad-hoc pipeline? First of all, it appears that the authors did not perform any kind of data curation to make sure that the new species did not include contaminations, deep paralogy or LGT issues, and in our opinion this is a must. Likewise, there is no information about the alignment algorithm and trimming (if performed).

Marker genes were searched in all taxa using the Phyla-AMPHORA pipeline. This pipeline includes a final trimming step using ZORRO (Wu, Chatterji, and Eisen 2012). The Phyla-AMPHORA marker gene set comprises ‘phylum-specific’ genes for the *Alphaproteobacteria* that are predominantly single-copy and have predominantly been vertically inherited as concluded by the congruency of their respective phylogenies (see Wang and Wu, 2013). In addition to this, single-gene phylogenies were built for each Phyla-AMPOHRA marker gene for the *Alphaproteobacteria* and visually inspected for potential cases of distant paralogues, contaminants or laterally transferred genes. Please see paragraph two of subsection “Dataset assembly (taxon and gene selection)” for a more detailed explanation.

5) Regarding phylogenetic analyses, it seems the LG replacement matrix was chosen without even comparing its better fit statistically (e.g. AIC or BIC). The authors use +R6 and +R8 to account for among-site rate heterogeneity in different analyses, but without an apparent reason for that. Lastly, please provide information on the ESS values for Bayesian runs to have a better grasp on the chain convergence.

We tested several models for their fit to our dataset that comprises 120 taxa and 200 conserved single-copy genes. We invariably found that the LG empirical amino acid replacement matrix fits the data better. This is the case for both simpler models that do not account for compositional heterogeneity across sites (see Table S5 in Supplementary file 2 and subsection “Phylogenetic inference”) as well as those that do (see Table S6 in Supplementary file 2).

With respect to accounting for rate variation across sites, the R model (a general rate variation model that includes the discretized Γ as a special case) is more flexible and tends to fit most datasets better than the G (Γ) model (see e.g., Soubrier et al., 2012, Mol. Biol. Evol. 29 (11)). This can be seen in the new Supplementary file 2, Table S7 which shows several fitting criteria (e.g., AIC, AICc, and BIC) for the best fitting model (i.e., LG+ES60+F) combined with different models to account for across-site rate heterogeneity (e.g., G, R4, R5 and R6). As Supplementary file 2, Table S7 shows, R6 increases the fit of the overall model to our dataset. This, however, came at the expense of significantly greater computational times.

For the phylogenetic analyses of the nucleotide alignment of 16S rRNA gene, we chose the GTR+F+R8 based on our previous knowledge that the R model with an increasing number of categories fits the data batter and because for this dataset increasing model complexity did not considerably increased computational time.

Please see the new Supplementary file 2, Table S8 for several summary statistics for each PhyloBayes MCMC chain.

6) One of aspects why the phylogeny of alphaproteobacteria is of broad interest is the mitochondrial lineage. We wonder why the authors did not try to place the mitochondria into their analyses. We assume this would bring additional biases into an already difficult dataset, but we think we could have gotten an interesting insight given the vast amount of analyses performed with various strategies to reduce systematic errors.

Our primary goal was to get a robust consensus phylogeny of the *Alphaproteobacteria*. The *Alphaproteobacteria* is ancient and diverse enough that inferring its phylogeny alone presents several challenges even when excluding mitochondria, clearly the most divergent and fast-evolving of its lineages. As the reviewers note, and as discussed in the Conclusions, incorporating mitochondria in our datasets will exacerbates the already strong biases in the data, and therefore represents additional sources of potential artefacts in phylogenetic inference. We are currently undertaking another study whose main goal is to phylogenetically place the mitochondrial lineage among an increasing diversity of alphaproteobacteria as we simultaneously attempt to ameliorate potential phylogenetic artefacts.

7) Abstract: it would be best to remove the fifth sentence, given that the support for these findings is not definitive. Additionally, it is important that you add that this study proposes an updated taxonomy for alphaproteobacteria, which is one of the major outcomes of your study.

We have removed the sentence in the Abstract that refers to the potential deep branching of the *Geminicoccaceae* as sister to all other free-living alphaproteobacteria. Additionally, as suggested by the reviewer, we have now added a sentence on the proposal of a synthetic higher-level taxonomy for the *Alphaproteobacteria*.

8) Conclusions: please remove the last two sentences of the conclusion. The one before last could be said for every study ever done. The last sentence is a bigger topic but including a single sentence in the conclusions fails to do it justice. If you want to discuss this issue, please include a paragraph in the discussion – as is, it comes out of the blue (and it's not clear why phylogenetic inference will be improved; if additional sampling keeps adding long branches, it may very well be that more uncertainty is introduced).

We have removed the two sentences and added a new one that best concludes the paragraph.

Introduction, third paragraph: It is generally well accepted that these three factors (few taxa, few genes, and models with poor fit) lead to systematic error. But your claim that previous studies were compromised by one or more of these factors in this section seems very hand-wavy. Can you give specific examples? Simply saying taxon sampling / model usage was poor in this or that study seems subjective – please give specific information as to why these studies had suboptimal designs (e.g., how many taxa were included, which of the major groups were sampled, why the model was a poor fit, etc.)

Further details about the shortcoming of some of these studies have been added.

Subsection “Compositional heterogeneity appears to be a major confounding factor affecting phylogenetic inference of the Alphaproteobacteria”, second paragraph: please briefly introduce in a short paragraph how you built the data matrix before you start describing how you analyzed it.

We have now briefly explained the nature and provenance of the 200-gene dataset, provided an appropriate reference and referred to the Materials and methods for more details.

Subsection “The Holosporales is unrelated to the Rickettsiales and is instead most likely derived within the Rhodospirillales” and “The Geminicoccaceae might be basal to all other free-living alphaproteobacteria (the Caulobacteridae)”: there are no page (or supplement size) restrictions, so please show the data.

We have now added two new figures that show these data. Please see Figure 3—figure supplement 7 and Figure 2—figure supplement 5.

Figure 3: the figure lists taxonomy family names (e.g., Holosporales) but the legend discusses order family names (e.g., Holosporaceae) and what the triangles correspond to is not explained. Please clearly annotate the figure.

We have amended the legend of Figure 3 accordingly. Collapsed clades have also now been labeled with taxon names for a much more straightforward interpretation of the figure.

Figures 2 and 3: the color-coding scheme of the different clades doesn't appear consistent. Please revise.

We have revised the figure colors and labelled taxa in Figure 3 for an easier interpretation.